# Neural signatures of indirect pathway activity during subthalamic stimulation in Parkinson's disease

Leon A. Steiner[1,2,3], David Crompton[1,4], Srdjan Sumarac [1,4], Artur Vetkas[1,5], Jürgen Germann [1,5,6], Maximilian Scherer[1,4], Maria Justich[1,7,8], Alexandre Boutet[9], Milos R. Popovic [4,10,11], Mojgan Hodaie [1,5,6,11,12], Suneil K. Kalia[1,5,6,10,11,12], Alfonso Fasano [1,7,8,11,12], William D. Hutchison WD[1,6,11,13], Andres M. Lozano [1,5,6,11,12], Milad Lankarany[1,4,11], Andrea A. Kühn [2] & Luka Milosevic [1,4,10,11,12] ✉

Deep brain stimulation (DBS) of the subthalamic nucleus (STN) produces an electrophysiological signature called evoked resonant neural activity (ERNA); a high-frequency oscillation that has been linked to treatment efficacy. However, the single-neuron and synaptic bases of ERNA are unsubstantiated. This study proposes that ERNA is a subcortical neuronal circuit signature of DBS-mediated engagement of the basal ganglia indirect pathway network. In people with Parkinson's disease, we: (i) showed that each peak of the ERNA waveform is associated with temporally-locked neuronal inhibition in the STN; (ii) characterized the temporal dynamics of ERNA; (iii) identified a putative mesocircuit architecture, embedded with empirically-derived synaptic dynamics, that is necessary for the emergence of ERNA in silico; (iv) localized ERNA to the dorsal STN in electrophysiological and normative anatomical space; (v) used patient-wise hotspot locations to assess spatial relevance of ERNA with respect to DBS outcome; and (vi) characterized the local fiber activation profile associated with the derived group-level ERNA hotspot.

Parkinson's disease (PD) is a common movement disorder that is associated with progressive degeneration of nigrostriatal dopaminergic projections, and has been associated with excessive oscillatory and neuronal synchronization of basal ganglia circuits[1]. It is now widely accepted that deep brain stimulation (DBS) of the subthalamic nucleus (STN) disrupts this pathological synchronization[2], but the neuronal and synaptic circuit underpinnings of this intervention are of ongoing debate[3].

The STN lies at a junction of cortical and pallidal projections, and the corresponding hyperdirect and indirect pathways have been

[1]Krembil Brain Institute, University Health Network, Toronto, ON M5T 1M8, Canada. [2]Department of Neurology, Charité-Universitätsmedizin Berlin, Berlin 10117, Germany. [3]Berlin Institute of Health (BIH), Berlin 10178, Germany. [4]Institute of Biomedical Engineering, University of Toronto, Toronto, ON M5S 3G9, Canada. [5]Division of Neurosurgery, Toronto Western Hospital, Toronto, ON M5T 2S8, Canada. [6]Department of Surgery, University of Toronto, Toronto, ON M5G 2C4, Canada. [7]Department of Neurology, University of Toronto, Toronto, ON M5S 3H2, Canada. [8]Edmond J. Safra Program in Parkinson's Disease, Morton and Gloria Shulman Movement Disorders Clinic, Toronto Western Hospital, Toronto, ON M5T 2S8, Canada. [9]Joint Department of Medical Imaging, University of Toronto, Toronto, ON M5G 1×6, Canada. [10]KITE Research Institute, University Health Network, Toronto, ON M5G 2A2, Canada. [11]Center for Advancing Neurotechnological Innovation to Application (CRANIA), Toronto, ON M5T 2S8, Canada. [12]Institute of Medical Sciences, University of Toronto, Toronto, ON M5S 1A8, Canada. [13]Department of Physiology, University of Toronto, Toronto, ON M5S 1A8, Canada. ✉e-mail: luka.milosevic@mail.utoronto.ca

implicated in both PD and the mechanism of action of STNDBS. Recent multi-modal work, that combined high resolution electrophysiology, state-of-the art imaging tools, and biophysical modeling, has been able to characterize the circuit signature of hyperdirect pathway activation in PD[4]. Furthermore, early optogenetic efforts to disentangle the subcircuit elements that account for the therapeutic effect of STN-DBS have pointed to antidromic stimulation of the motor cortex via the hyperdirect pathway[5,6]. However, more recent observations that antidromic activation of cortical neurons by STN-DBS wane in the time course of minutes despite enduring therapeutic effects have cast doubt on the causal nature of this phenomenon[7]. Moreover, stimulation of the globus pallidus internus (GPi) is not associated with hyperdirect pathway activation, despite producing similar therapeutic efficacy to STN-DBS[7].

The role of the indirect pathway has received attention as well. Within the STN, DBS has been shown to suppress neuronal firing[6,8] by means of persistent synaptic activation of GABAergic inputs from the external pallidum (GPe)[9,10]; thus supressing pathological bursting activity in the STN that has been associated with the dopamine-depleted state[1,11]. At the same time, single neuron spiking patterns in both the external and internal parts of the pallidum have been shown to be driven and "regularized" by STN-DBS, potentially mediated by recruitment of excitatory STN efferents that widely project to the pallidum[12–14]. The discrepancy of STN somatic inhibition and efferent entrainment has been previously conceptualized as the so-called soma-axon decoupling theory[15]. However, STN-DBS can also be expected to elicit antidromic activations of the GPe projections to STN, an aspect of the DBS mechanism of action that has hitherto received little attention. Considering that STN-projecting GPe neurons give rise to extensive local axon collaterals[16], STN-DBS can be expected to result in the recruitment of these collateral projections[17] and release of neurotransmitters at synaptic terminals. Thus, synaptic inhibition of the GPe by recruited axon collaterals may compete with incoming excitatory input via the STN efferents. An integrative account of these synaptic subcortical circuit activations is missing, which motivated the work presented is this study.

Recently, it has been suggested that the reciprocal STN-GPe connection might give rise to evoked potentials of an oscillatory nature that have been termed "evoked resonant neural activity" (ERNA)[18]. ERNA amplitude has been shown to correlate with clinical improvement during STN-DBS[18,19]. Importantly, unlike hyperdirect pathway activation, it has recently been shown that ERNA is present during therapeutically effective GPi-DBS[20]. Given that both STN and GPi are intimately connected to GPe, it seems plausible that ERNA may represent a convergent subcortical signature that is therapeutically linked to the engagement of the indirect pathway[21]. However, the neural origin of ERNA has been questioned previously[22] and to date, single-neuron substrates of ERNA have not been characterized in detail[23,24].

In the present study, we employed a multi-modal approach to scrutinize ERNA as a physiological signature that may represent a neural substrate of the human indirect pathway (Fig. 1)[25]. Specifically, we leveraged intraoperative recordings to establish a single-neuron correlate of the ERNA waveform in patients undergoing DBS surgery for PD. Moreover, we validated our empirical findings within the context of a computational model, which allowed for detailed characterization of the specific circuit architecture and synaptic dynamics that are necessary for the emergence/development of ERNA in silico. Next, we performed high-resolution electrophysiological mapping to probe the spatial topology of ERNA, and performed correlative spatial analyses to investigate the importance of this signature towards therapeutic benefit achieved by STN-DBS. Lastly, we characterized the fiber activation profile in the vicinity of the ERNA hotspot using a recently derived subcortical fiber atlas[26]. Ultimately, these multi-modal results elucidate putative synaptic and circuit properties required to produce ERNA, highlighting the importance of indirect pathway activation for its manifestation.

## Results

### Relationship between the ERNA waveform and STN spiking

To examine the relationship between the ERNA waveform and single neuron spiking, we analyzed microelectrode data in which single-unit STN activity was captured concurrent to ERNA, during stimulation across various intensities. For each recording site, we constructed interstimulus waveform averages and spike histograms from stimulation trains at 100 Hz and stimulation intensities of 30, 50, and, 100 μA (e.g., Fig 2A).

Increased stimulation current amplitudes resulted in reduced neuronal firing during high-frequency stimulation (HFS) in conjunction with increased amplitudes of ERNA waveforms. Patterns of both features were preserved throughout stimulation intensities (Fig. 2B, C). To quantify the relationship between ERNA waveform shape and neuronal suppression, a set of linear mixed-effects regression models was constructed. First, a linear mixed-effects regression model demonstrated a significant relationship between the log-transformed amplitude of the ERNA waveform and neuronal firing, both extracted from a matched temporal window that captured the oscillatory component of the ERNA waveform (Fig. 2D, left panel; estimate of fixed slope: $-13.069 \pm 2.217$, $\text{tstat}_{1,58}$: $-6.143$, $n = 60$, $p = 3.167e-7$). Next, a separate linear mixed-effects regression model showed that the log-transformed amplitude of the second peak (P2) was significantly related to the amplitude of the trough in the patterned interstimulus spiking (Fig. 2E, left panel; estimate of fixed slope: $-7.361 \pm 1.813$, $\text{tstat}_{1,58}$: $-4.06$, $n = 60$, $p = 2.503e-11$). These data demonstrate a relationship between the shape of the ERNA waveform and similarly patterned neuronal suppression in the interstimulus interval (inhibition of neuronal activity that is time locked to peaks of the ERNA waveform, whereby the strength of the neuronal inhibition is proportional to the size of the respective ERNA peak), across recording sites. In a subsequent analysis, relationships between the ERNA waveform and patterned neuronal

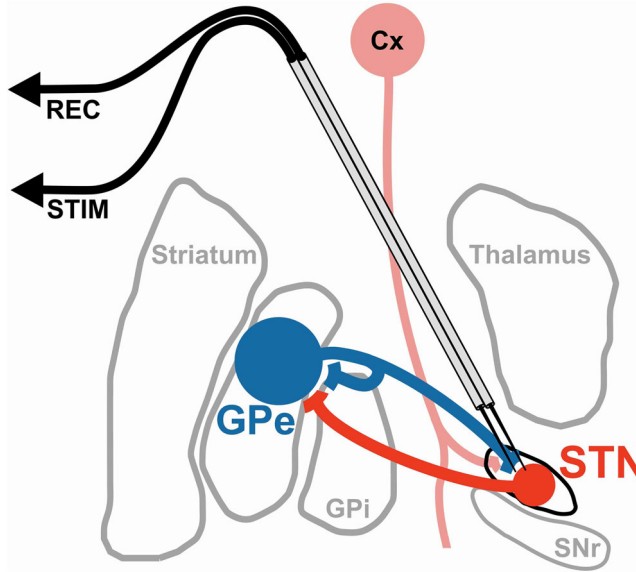

**Fig. 1 | Stimulation and recording montage.** Schematic of two closely spaced microelectrodes that were advanced into the STN of patients to record evoked potentials and neuronal activity. Throughout the analyses, neural activity was recorded with one microelectrode (REC) during stimulation trains from the adjacent microelectrode (STIM). STN subthalamic nucleus, GPe/i globus pallidus externus/internus, SNr substantia nigra pars reticulata, Cx cortex.

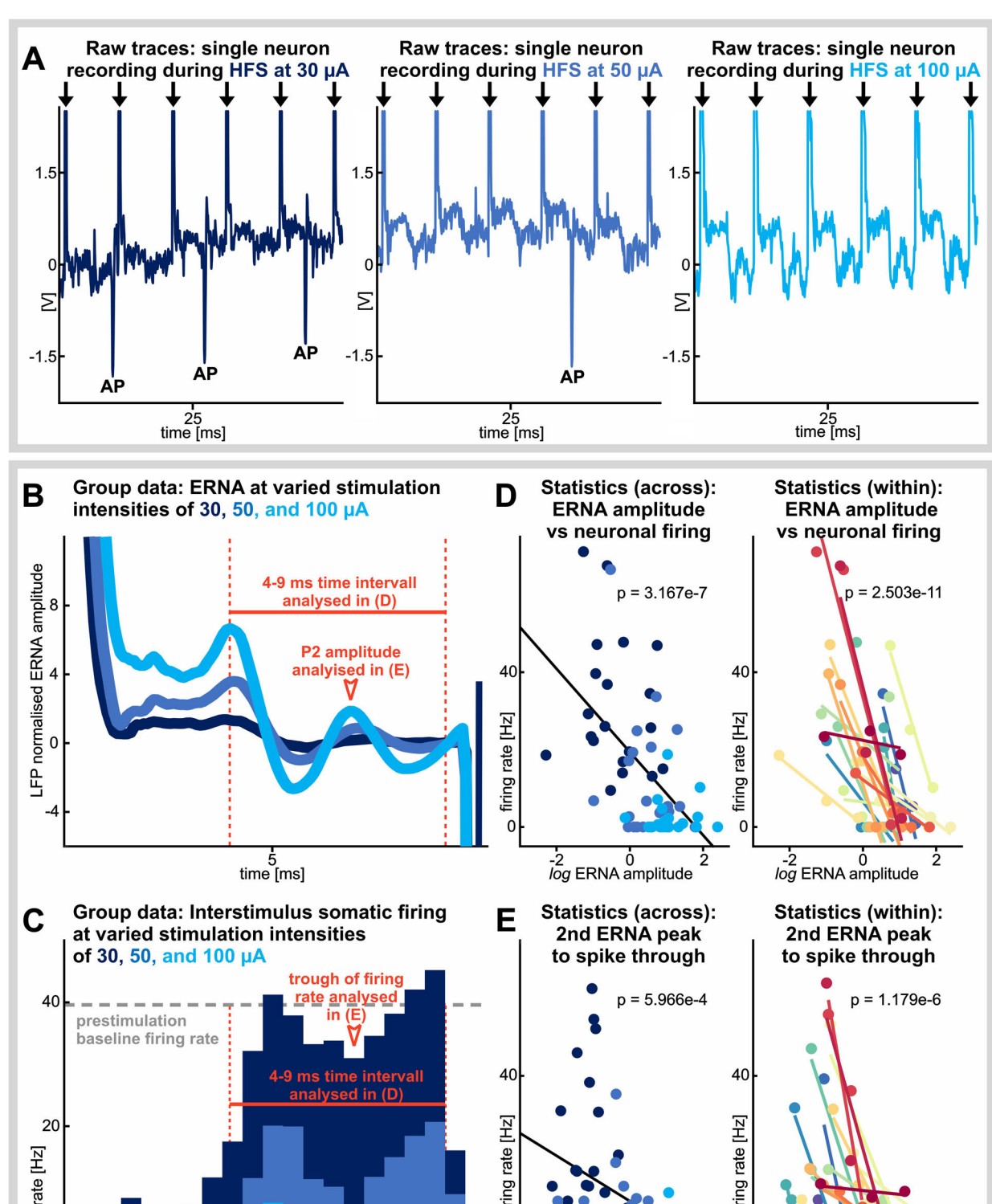

suppression were tested within recording sites, across the three stimulation intensities. This within-neuron analysis confirmed that a greater ERNA waveform amplitude was linked to a stronger suppression of neuronal firing for individual cells (Fig. 2D, right panel; tstat$_{1,19}$: −14.891, n = 20, p = 5.966e-4) and that a larger P2 amplitude was associated with a deeper trough in interstimulus spiking (Fig. 2E, right panel; tstat$_{1,19}$: −7.897, n = 20, p = 1.179e-6).

Together, the presented analyses suggest that the ERNA waveform shape is tightly linked to the pattern of inhibition of neuronal spiking in the STN.

### Temporal dynamics of the ERNA waveform

To study the evolution of the ERNA waveform over time (e.g., Fig 3A), we analyzed the dynamics of the first (P1) and second (P2) ERNA peaks.

**Fig. 2 | The evoked resonant neural activity (ERNA) waveform shape is linked to patterned inhibition of STN single neuron spiking. A** Representative traces of ERNA and action potential (AP) spiking at different stimulation intensities, all at high frequency stimulation (HFS; 100 Hz). **B** Interstimulus waveform averages (*n* = 20 recording locations). **C** Interstimulus spike histogram averages from the same locations at the same stimulation intensities, all at 100 Hz. Dashed horizontal line illustrates pre-stimulation average firing rate. **D** Scatter plots illustrating the relationship between the amplitude of ERNA and degree of neuronal inhibition across (left; *p* = 3.167e-7) and within neurons (right; *p* = 2.503e-11). **E** Scatter plots

illustrating the relationship between the amplitude of the second ERNA peak (P2) and the local minimum in the interstimulus histogram across (left; *p* = 5.966e-4) and within neurons (right; *p* = 1.179e-6). *P* values for across neuron comparisons are taken from separate linear mixed-effects models (**D**, **E** left panels). For within neuron comparisons, Rho values are taken from Spearman's correlations for each neuron and compared against 0 using a 1-sample, 2-tailed *t* test to arrive at the *p* values shown (**D**, **E** right panels). *P* values are corrected for multiple comparisons using the Bonferroni correction (4 hypotheses).

To the test the frequency-specificity, waveform dynamics were visualized for both high (100 Hz; Fig. 3B) and low (10, 20, 30, 50 Hz; Supplementary Fig. 1) stimulation frequencies. Changes to the dynamics of P1 and P2 could be observed at high, but not low stimulation frequencies. Therefore, the subsequent analysis focused only on the dynamics of P1 and P2 during HFS. 100 Hz stimulation resulted in depression of P1 (mean of ratio $P1_{40-45th}/P1_{first}$ = 0.765, $tstat_{1,11}$: 21.194, $n$ = 12, $p$ = 2.866e-10) which was contrasted by the growing amplitude of P2 (mean of ratio $P1_{40-45}/P1_{first}$ = 1.395, $tstat_{1,11}$: 6.882, $n$ = 12, $p$ = 2.651e-05). To visualize dynamics of P1 and P2 at the group level, a plot of the average amplitudes was constructed (Fig. 3B). Finally, dynamics of P1 were correlated with dynamics of P2, demonstrating a statistical relationship between the depression of P1 and growth of P2 (Fig. 3C; Spearman correlation, $n$ = 12, Rho = −0.6853, $p$ = 0.0173).

These results suggest opposing dynamics of individual peaks of the ERNA waveform. The degree of depression of P1 was linked to the dynamic emergence of P2. These empirical findings led to the development of a conceptual framework to explain the synaptic origin of ERNA (Fig. 3D).

## Computational interrogation of ERNA circuit architecture & dynamics

To recapitulate the empiric findings in the context of the engaged mesocircuit, a biophysical model was constructed based on the described conceptual framework (details on model construction are available in the computational Methods sub-section). We constrained model parameters to emulate a viable anatomical architecture (Fig. 4A) and empirical findings of the short-term synaptic depression dynamics at individual synapses of the cortex-STN-GPe mesocircuit (Fig. 4B).

Similar to experimental results, the growth of P2 in the efferent GPe activity approximating ERNA only developed with HFS (Fig. 4Ci), during which synapse-specific short-term synaptic dynamics were considered. Low frequency stimulation, during which short-term synaptic depression does not occur, was not able to reproduce the evolutionary dynamics of ERNA (Fig. 4Cii). This is because the co-activation of inhibitory and excitatory inputs to GPe would indefinitely cancel each other out, leading to a lack of recurrent inhibition to STN that is necessary to produce P2 dynamics. These computational results corroborate our empirical findings with low frequency stimulation which show a lack of P2 development with low frequency stimulation (Supplementary Fig. 1).

With respect to the STN efferent activation, ERNA dynamics were only reproducible when the efferent STN synapse was modeled to be static (i.e., lack of short-term synaptic depression); which is suggested in several experimental works[12,27]. Implementation of a rapidly depressing synapse was unable to reproduce the ERNA dynamic (Fig. 4Ciii) since a depressing excitation from STN would no longer sufficiently recruit GPe-mediated feedback inhibition. These results emphasize the importance of synapse-specific dynamics of short-term synaptic plasticity; wherein, cortico-STN synapses rapidly depress[10], GPe-STN and GPe-GPe synapses depress to some degree but are resilient thereafter[9,10], and STN efferent synapses are resilient[12,27] (Fig. 4B).

Lastly, when intrinsic GPe-GPe connectivity was limited, the model was incapable of reproducing the ERNA dynamic (Fig. 4Civ). The lack

of recurrent intrinsic connectivity resulted in a net decrease in inhibitory conductivity in the GPe network during simulation, whereas simulations with a higher degree of recurrent connectivity had greater net inhibitory conductance. These results suggest that the development of P2 in STN is dependent upon a synaptic competition of inhibitory and excitatory inputs to GPe; whereby the efficacy of the GPe-GPe projection slowly decays while the STN-GPe synapse remains static. The resultant net excitation of GPe that occurs over successive stimuli thereafter feeds-forward to produce an increasing recurrent inhibition STN manifested as a growing P2.

Overall, the modeling results suggest that the synaptic origin of ERNA is dependent upon HFS, which elicits synapse-specific dynamics of short-term synaptic plasticity, and a viable anatomical architecture is necessary in which GPe-STN fibers include GPe-GPe axon collaterals that are invaded and thereby activated by STN-DBS (in addition to the activation of local STN inputs and outputs).

## Structuro-functional topology and clinical relevance of ERNA

Figure 5A illustrates a representative ERNA waveform average topology from a single surgical trajectory. At the group level (Fig. 5B), the ERNA waveform began to emerge in the vicinity of the dorsal STN border, was of largest amplitude ~1.5 mm into the STN, and slowly decayed as the microelectrodes were advanced more ventrally.

In a subsequent step, electrode trajectories were localized in a normalized anatomical space (MNI-space) and ERNA amplitudes were reconstructed in the form a 3D heatmap relative to an atlas-based STN model (Fig. 5C). The group-level ERNA hotspot is shown to be in the vicinity of previously described DBS clinical sweetspot[28]. Improvement in the Unified Parkinson's Disease Rating Scale part III (UPDRSIII) total score and akinetic-rigid subscore correlated with patient-specific ERNA hotspot proximity to the DBS clinical sweetspot (Fig. 5D; UPDRSIII, Rho = −0.5947, $p$ = 0.0168; akinetic-rigid, Rho = −0.6058; $p$ = 0.0120). Fibers crossing through the group level ERNA hotspot (1 mm radius) consisted of 30% GPe-STN, 30% STN-GPe, 12% supplementary motor area (SMA)-STN, 10% premotor-STN, and 19% primary motor cortex (M1)-STN fibers (Fig. 5E, F).

These results highlight the spatial topology of ERNA, reflect the importance of its proximity to the DBS sweet spot for producing postoperative clinical improvement, and demonstrate the predominance of STN-GPe/GPe-STN fiber activations.

## Discussion

In patients with PD undergoing DBS surgery, we employed a multi-modal approach to characterize the neuronal subcortical circuit signature linked to the therapeutic potential of subthalamic DBS at the synaptic level. Specifically, we were able to (i) show that each peak of the ERNA waveform is associated with temporally-locked neuronal inhibition in the STN, (ii) corroborate the temporal dynamics of ERNA by way of projection-specific short-term synaptic dynamics within the recurrent GPe-STN loop, (iii) identify a putative mesocircuit architecture, embedded with empirically-derived synaptic dynamics, that is necessary for the emergence of ERNA in silico, (iv) localize ERNA to the dorsal STN in electrophysiological and normative anatomical space, (v) predict postoperative clinical improvement from the distance of the patient-specific ERNA to the previously published DBS sweet spot,

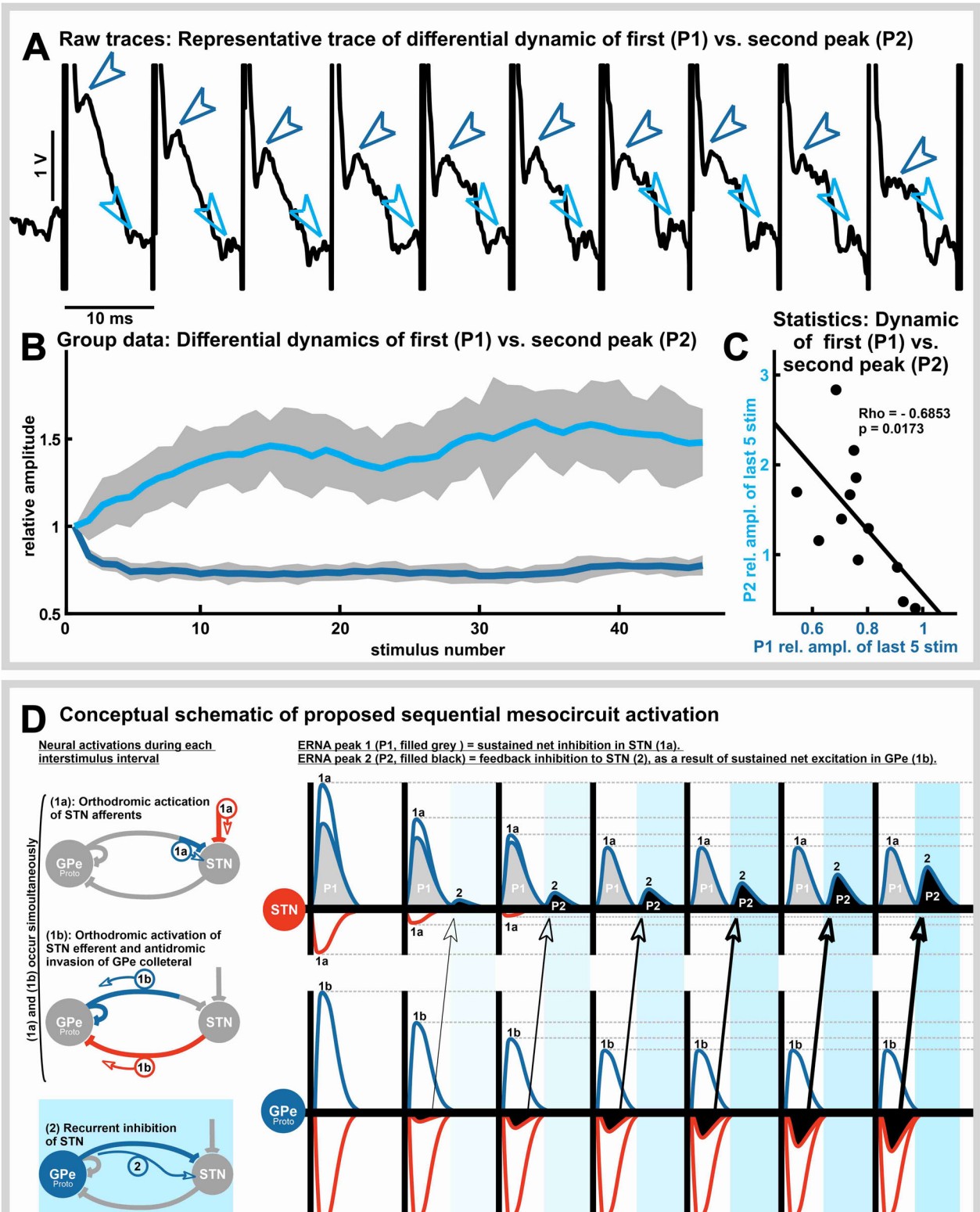

**A Raw traces: Representative trace of differential dynamic of first (P1) vs. second peak (P2)**

**B Group data: Differential dynamics of first (P1) vs. second peak (P2)**

**C Statistics: Dynamic of first (P1) vs. second peak (P2)**

Rho = - 0.6853
p = 0.0173

**D Conceptual schematic of proposed sequential mesocircuit activation**

Neural activations during each interstimulus interval

ERNA peak 1 (P1, filled grey ) = sustained net inhibition in STN (1a).
ERNA peak 2 (P2, filled black) = feedback inhibition to STN (2), as a result of sustained net excitation in GPe (1b).

(1a): Orthodromic activation of STN afferents

(1b): Orthodromic activation of STN efferent and antidromic invasion of GPe colleteral

(2) Recurrent inhibition of STN

and (vi) characterize the putative local fiber activation profile associated with the derived ERNA hotspot.

Experiments using DBS macrocontact recordings have previously demonstrated that DBS evoked potentials can be observed throughout the basal ganglia, and computational work has suggested that ERNA may arise from the reciprocal connections between the STN and GPe[25].

Further, activation of STN efferents has been shown to produce patterned neuronal excitation in the primate GPe[12], which is in fact a phase-shifted representation of the patterned inhibition shown herein. Kita et al. have shown that synaptically-mediated neuronal excitations in GPe during ongoing STN-HFS progressively increase in strength within 10 pulses of 100 Hz[14] and are thus in good temporal alignment

**Fig. 3 | Temporal evolution may reflect synaptic dynamics of ERNA. A** Example trace of microelectrode recording in the STN illustrating that the depression of inhibitory inputs during the first 10 pulses for the first peak (P1; dark blue arrows) is associated with an increase in amplitude of a second, resonant evoked field (P2; light blue arrows). **B** Group data (n = 12.) of differential dynamics of P1 (dark blue) and P2 (light blue) across successive stimuli; y-axis depicts P1 and P2 peak amplitudes relative to the first respective peak in the stimulus train. Shade represents standard error of the mean. **C** Scatter plot illustrating the relationship between the dynamics of the decaying P1 (x-axis) and increasing P2 (y-axis); dynamics were measured as $P1_{40\text{-}45th} / P1_{first}$ and $P2_{40\text{-}45th} / P2_{first}$. Rho and p value are taken from Spearman's correlation (Rho = −0.6853; p = 0.0173). **D** Conceptual schematic to illustrate how STN stimulation may trigger a cascade of synaptic events that gives rise to ERNA. Left: Schematic representation of activated fibers at two timepoints (1a/b and 2) during a single interstimulus interval. Note that (1a) and (1b) are expected to occur simultaneously within a "monosynaptic" time course in direct response to individual stimuli, whereas (2) occurs subsequently/consequently, at a "disynaptic" time course (depicted by blue shading). Right: These synaptic responses change in amplitude across successive interstimulus intervals as a result of short-term synaptic plasticity. Vertical lines represent individual stimuli

at HFS. Blue positive-going evoked field potentials correspond to inhibitory afferent activations, whereas negative-going red evoked field potentials represent activations of excitatory inputs. (1a) depicts direct simultaneous activations of the inputs to STN. Sustained GPe transmission is paired with rapidly depressing cortical transmission, which leads to a sustained net inhibition in STN (this inhibits STN spike firing and produces a hyperpolarization of the membrane potential which is reflected as P1 of STN ERNA, filled gray). (1b) depicts direct activation of STN efferent outputs (even though spike firing is inhibited by GPe activation, STN efferent axons are nevertheless expected to be activated by DBS pulses) and GPe-GPe collaterals (by way of antidromic activation of GPe-STN projections, and subsequent invasion of GPe-GPe collaterals). Sustained GPe transmission paired with even more sustained STN transmission leads to an increasing net excitation in GPe. (2) depicts that the increasing net excitation in GPe feeds back to the STN, producing recurrent inhibition of STN (this produces a second hyperpolarization of the membrane potential in STN within the interstimulus interval, which is reflected as P2 of STN ERNA, filled black). Effectively, the recurring activations of GPe inhibit the generation of STN spike firing, while contributing to the positive-going voltage peak deflections in the ERNA waveform (i.e., due to a loss of negatively-charged ions in the extracellular field potential recordings due to GABAergic activations).

with the emergence of recurrent neuronal inhibition presented in our study (Figs. 3, 4, and Supplementary Fig. 1). Collectively, these works provide electrophysiological support for the engagement of the reciprocal inhibitory-excitatory GPe-STN mesocircuit network. To date however, single-neuron spiking has not previously been reported in recordings that simultaneously capture the ERNA waveform, as ERNA has only been studied in DBS macroelectrode contexts. Our finding that the ERNA waveform peaks are temporally locked to inhibitions of STN single-neuron activity substantiates the hypothesis that ERNA is likely a GPe-mediated signature (given that GPe is the major source of inhibition to STN). Accordingly, the persistent hyperpolarizing tone that is produced by ERNA may underly the well-known rebound burst phenomenon that occurs after prolonged hyperpolarization; mediated by deinactivation of T-type calcium channels[29]. This is consistent with observations of rebound bursts in STN that occur after stimulation cessation[8] (shown in both in our empirical and in silico data; Supplementary Fig. 2).

Importantly, our in silico work suggests that ERNA depends on the activation of the STN-GPe circuit. While the initial peak of the ERNA waveform can be explained by sustained GPe-mediated inhibition to STN, the second peak is proposed to be the result of synaptic competition at the level of GPe that is dominated by STN efferent activation[14], which ultimately feeds back to STN to produce recurrent inhibition. This dependence on an intact reciprocal network is corroborated by findings that ERNA is not produced by DBS-like stimulation of STN in acutely isolated rat brain slices that do not contain GPe neurons, but only GABAergic axonal blebs[9]. This suggests that the integrity of GPe-STN circuit is necessary to produce the phenomenon of recurrent inhibition.

Through widely arborized axon collaterals, single STN neurons project to the GPe (and to both basal ganglia output structures, GPi and SNr)[30]. Additionally, GPe neurons send axonal projections to the STN (that also collateralize in GPi and SNr)[31]. However, GPe neurons furthermore give rise to local axon-collaterals[16,32] that result in functionally relevant synaptic inhibition to neighboring GPe-neurons[33], unlike STN neurons which appear not to have mutual synaptic connectivity[10,34]. These anatomical insights were indeed taken into consideration in the development of the mesocircuit architecture of our computational model, in which DBS impulses directly activated inputs to STN, STN efferent synapses to GPe, as well as GPe-GPe synapses via invasion[17] of axon collaterals. At the level of STN, direct afferent input activation would cause competition of synaptic inputs, which is dominated by inhibitory inputs in terms of quantity[35] and resiliency[9,10] of activated fibers, producing the first peak of the ERNA waveform. Synaptic competition would also simultaneously occur at

the level of GPe. In this context however, the static nature of STN efferents, coupled with a slowly decaying inhibitory synaptic transmission, produces a progressively growing net excitatory response, which subsequently leads to feedforward recurrent inhibition to STN, producing the resonant peak of the ERNA waveform. The inhibitory nature of the STN ERNA waveform is corroborated by temporally locked patterned inhibition shown here; whereas temporally locked net excitations in GPe during STN-DBS have been shown in primate experiments[12].

While convergence of indirect and hyperdirect pathway fibers at the level of STN has been shown to be instrumental in the orchestration of pathological subcortical oscillatory activity[1,4] and neuronal synchrony[36], there has been debate about the relative contributions of engaged pathways to the therapeutic effect of subthalamic DBS[6,37]. However, beyond structural fiber activation profiles, we suggest that projection-specific synaptic dynamics have to taken into consideration to appreciate the functional consequences of such fiber activations (e.g., cortical suppression paired with GPe entrainment). Previous canonical optogenetic work has shown that high-frequency activation of hyperdirect pathway fibers produced antiparkinsonian effects[6]. However, in looking at the neuronal activity during this intervention, one can observe that high frequency activations actually produced inhibition of STN activity (while only low frequency activations produced excitatory responses)[6]. This can likely be explained by the fact that high-frequency activation leads to a suppression of the excitatory cortico-subthalamic drive, which has been corroborated by electrophysiological work in rodent subthalamic slices[10]. Because of the rapid depression of hyperdirect inputs during HFS, the effect of hyperdirect pathway stimulation on the generation of ERNA can be considered negligible. This is corroborated by the in silico findings presented in this study, which show similar dynamics of ERNA in the absence of hyperdirect inputs as compared to rapidly depressing hyperdirect inputs (Supplementary Fig. 3). Moreover, subsequent optogenetic works have shown that direct somatic inhibition of STN also produced antiparkinsonian benefits[38], and that activation of populations of GPe neurons that selectively project to STN could also produce long-lasting antiparkinsonian benefits (whereas global activation of GPe did not)[39]. As such, each of these optogenetics works may in fact support the importance of the indirect pathway in mediating antiparkinsonian effects.

Additionally, while activation (or, inactivation) of hyperdirect pathway fibers may be sufficient for producing antiparkinsonian effects during STN stimulation[6], it is a phenomenon that has been shown experimentally to not occur during therapeutically effective stimulation of the GPi[7]; whereas electrophysiological findings have emerged which suggest the presence of ERNA during effective GPi-

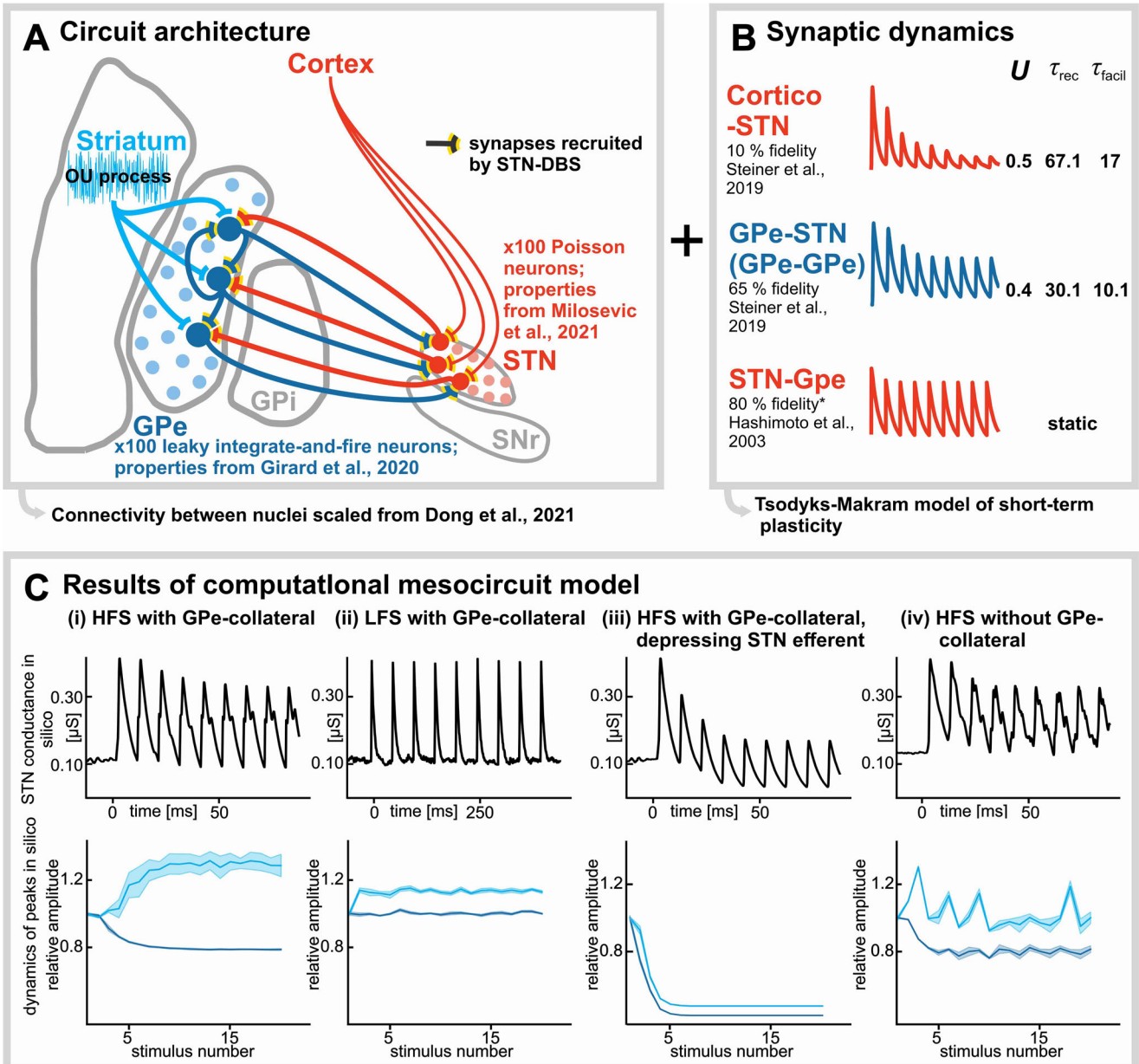

**Fig. 4 | Computational interrogation of the mesocircuit architecture and dynamics underlying emergence of the ERNA waveform. A** Schematic of the mesocircuit architecture which was able to emulate ERNA based on **B** DBS-induced depression of cortico-STN inputs (10% fidelity), persistent activation of GPe-STN inputs and recruited GPe-GPe axon collaterals (65% fidelity), and STN-GPe efferents (80% fidelity); OU = Ornstein–Uhlenbeck; $U$ = utilization constant; $\tau_{facil}$ = facilitation constant; $\tau_{rec}$ = recovery constant. **C** Modeling results showing the emergence of ERNA when (i) the aforementioned mesocircuit properties are considered, but lack of ability to reproduce the ERNA waveform (ii) during low frequency stimulation (LFS; i.e., when short-term synaptic dynamics are not considered), (iii) with depressing STN efferent fidelity, and (iv) in the absence of intrinsic GPe-GPe connectivity; highlighting the importance of each of these considerations. Upper panels: STN conductances in response to HFS. Lower panels: In silico dynamics of first (dark blue) and second (light blue) ERNA peaks; shade represents standard deviation of the mean; all plots depict the average of ($n = 10$) iterations of the model output.

DBS[20]. As described above, both STN and GPe neurons have extensive axonal branching which includes axon collaterals that terminate in GPi. As such, GPi DBS may also invade the reciprocal STN-GPe mesocircuit network which underlies the generation of ERNA[21]. In GPi, the initial ERNA peak is likely a result of direct activation of afferent inputs (producing a net hyperpolarization, due to the greater abundance of GPe inputs compared to STN). Through concurrent antidromic activation of GPe and STN afferents, and invasion of axon collaterals, GPi-DBS can produce STN-mediated release of glutamate in GPe, and GPe-mediated release of GABA in STN. The excitation of GPe would thereafter lead to recurrent inhibition of GPi, and therefore, the second peak of the ERNA waveform. A greater relative innervation of inhibitory inputs from GPe-to-STN as compared to GPe-to-GPi (which also receives inhibitory inputs from other sources; namely, striatum) may explain observations of greater ERNA amplitudes in STN compared to GPi[20].

Collectively, these experimental studies support the important role of indirect pathway[21], but not hyperdirect pathway[7], activation as an electrophysiological signature common to therapeutically effective STN- and GPi-DBS. It is perhaps this resonant subcortical mesocircuit phenomenon, which promotes GPe-mediated downstream inhibition throughout the broader basal ganglia network, that also underlies the slower time course anticorrelation observed within the convergent functional connectomic profile associated with both of these

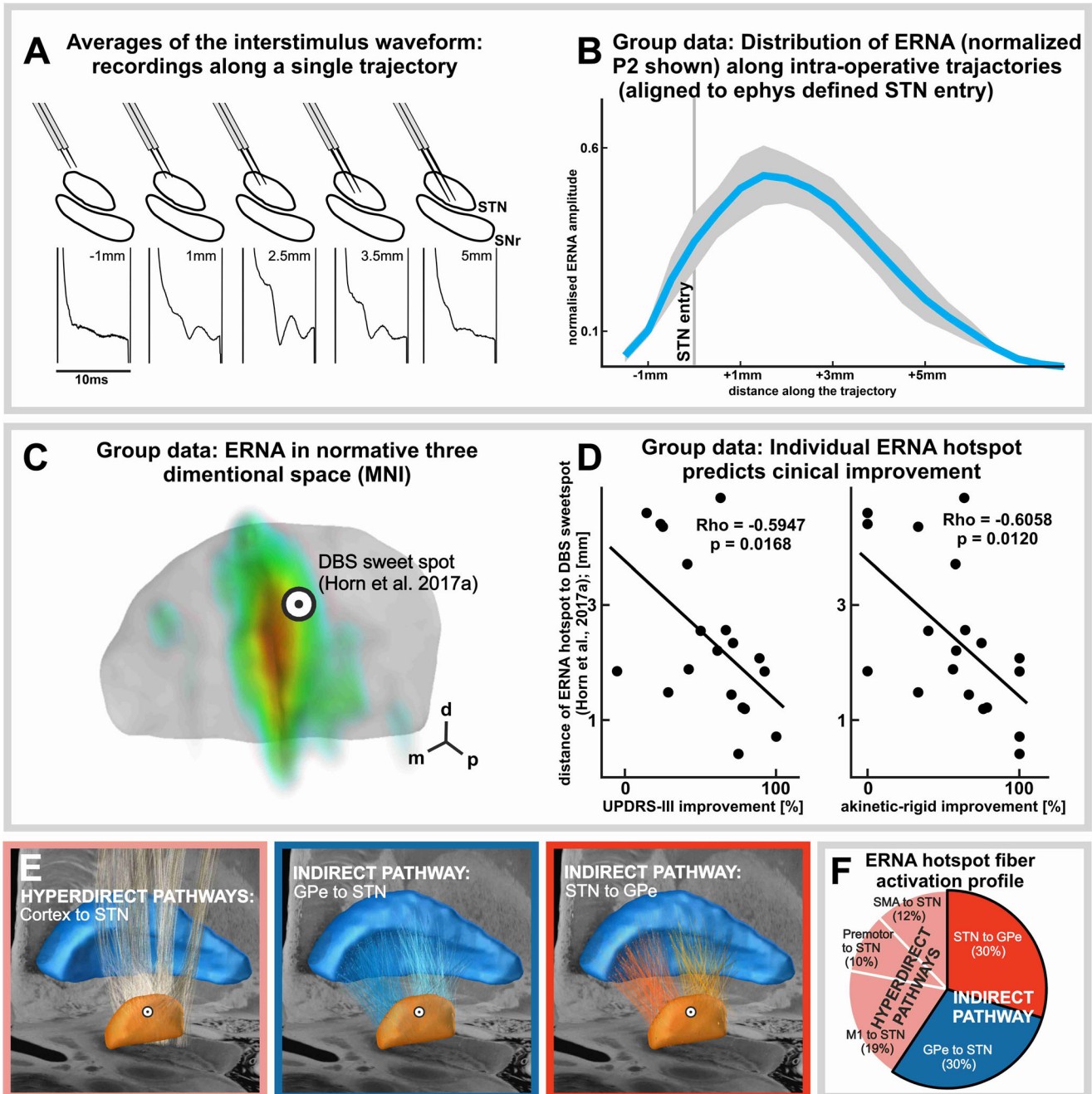

**Fig. 5 | Topographic mapping identifies ERNA as spatially defined electro-physiological signature of the indirect pathway that predicts postoperative clinical improvement. A** Individual examples of waveform averages along a representative trajectory. **B** Group data (*n* = 20) representation of the amplitudes of second peak (P2) normalized to the trajectory-specific maximum, mapped with respect to the electrophysiologically defined entry to STN based on intraoperative microelectrode recordings. X-axis refers to distance from STN entry. Shade represents standard error of the mean. **C** Group data (*n* = 19; postoperative neuroimaging was unavailable for one patient) of the ERNA amplitude heatmap in MNI space (superimposed onto an STN atlas rendering[59]); orientation arrows d = dorsal; m = medial; p = posterior. **D** Correlations of improvements in lateralized UPDRS III

(left; Rho = −0.5947; *p* = 0.0168) and akinetic-rigid (right; Rho = −0.6058; *p* = 0.0120) subscores and the Euclidean distance between the DBS clinical sweet spot[28] and the patient-specific ERNA hotspot. Scatter plots depict Spearman's correlations; *p* values were adjusted for multiple comparisons using the Bonferroni correction (2 hypotheses). **E** Renderings of primary motor cortex (M1; face, upper extremity, and lower extremity), supplementary motor area (SMA), and premotor hyperdirect pathway fibers, and GPe-STN and STN-GPe indirect pathway fibers based on the subcortical Petersen fiber atlas[26] (superimposed onto the DISTAL atlas STN and GPe renderings[59] and 7T brain scan in MNI 152 space[60]) along with **F** the fiber recruitment profile for the group level ERNA hotspot considering a 1 mm radius (see methods for further specifications).

interventions[40]. This subcortical network activation would also explain observations of ERNA waveforms in GPi during STN-DBS[25], as well as the inhibition of neuronal firing in STN during GPi-DBS[41].

Ultimately, STN-DBS may represent a double-edged intervention that not only decorrelates cortico-subthalamic connectivity, but also interacts with the subcortical circuit that has been suggested to

underly pathologically enhanced beta oscillations; namely, the local microcircuit in GPe and reciprocal GPe-STN connections[4,42,43]. Empiric evidence for this parallelism comes from the fact that beta power amplitude in the stimulation OFF period correlates with several ERNA features, most importantly its amplitude and frequency[23,25,44]. Furthermore, the high resolution electrophysiological mapping presented

in this study, showing that highest amplitude ERNA is localized to dorsal STN, is in spatial agreement with previous data sampled from DBS macrocontacts[18], localizations of beta power within the STN[45], and therapeutic sweet spots of subthalamic DBS[28,46]. These observations collectively suggest that subthalamic DBS likely engages the same subcortical mesocircuit network that gives rise to beta oscillations, imposing stimulation-induced suppression of pathological activity through high frequency driving of GPe-mediated inhibition (i.e., ERNA) in tandem with suppression of cortical influence on STN. ERNA can also be elicited by STN-DBS in dystonia, substantiating that ERNA may be reflective of a synapse-specific circuit-phenomenon rather than being disease specific[47].

Moreover, targeting of the subcortical network that gives rise to ERNA may enable further optimization of STN-DBS for therapeutic purposes. Given that the activation of GPe neuronal populations that preferentially project to STN can lead to long-lasting therapeutic benefits that persist beyond stimulation cessation[39], our findings may provide a means of selectively targeting these therapeutically-relevant GPe fibers. The potential to produce long-lasting benefits may have important implications in the context of closed-loop STN DBS implementations with frequent OFF stimulation periods. Scrutinizing synaptic signatures of neuronal circuit activation in humans thus offers unique translational potential to improve DBS stimulation paradigms and achieve therapeutic restoration of circuit function; particularly when employed in combination with the characterization of fiber recruitment profiles, as employed here and previously[48], that may enable circuit-specific interventions[39,49].

Whenever possible, the physiological components of our computational modeling framework utilized parameters to emulate synaptic dynamics based on empirical data in the present and previous studies. However, to the best of our knowledge, no microelectrode recording data exists on the dynamics of evoked potentials and corresponding firing in GPe in response to STN stimulation. Because pallidal firing is continuously driven by STN-HFS[12,14], we assumed a resilient STN output synapse. When we changed the STN output synapse to not be resilient in silico, the characteristic ERNA waveform did not develop (Fig. 4C). Indeed, this resiliency is supported by intraoperative biochemical studies reporting sustained glutamate levels in SNr during STN-DBS[27]. Future high-resolution microelectrode recording studies may provide further empirical support for the neuronal resiliency. Similarly, we did not obtain empirical data on the dynamics of the synaptic depression at the recruited GPe-GPe collateral, but we assumed similar dynamics as observed for direct afferent activation in STN[9,10]. Incomplete synaptic depression of local GABAergic axon collaterals in the GPe has also previously been reported in the rat[33]. Future simultaneous patch-clamp recordings of GPe neurons during STN stimulation in acute pallidal-subthalamic slices may provide more direct evidence of cell-type specific neuronal and synaptic recruitments and their involvement in the emergence of ERNA.

The use of single compartment abstract models to reproduce ERNA, while limited, does seem to affirm the theory of STN-GPe circuit engagement of previous work[25]. The firing patterns are only fit to have matched the mean firing rate described in previous works[12,35], while previous models of firing rates during DBS were more realistic due to the use of multi-compartment models that encapsulate the physiology more accurately. Unlike previous works though, we integrated short-term plasticity dynamics which seems to be sufficient to explain the temporal dynamics of ERNA, however slow/long term plasticity as was not included in this model, and the model does not account for dynamics that may occur over years of chronic stimulation, as applied clinically. The use of single compartment models is also limited in the reproduction of local field potentials, whereas a multi-compartment model would capture the intricacy embedded in the morphology of the tissue. However, the synaptic model is sufficient to produce the primary features of the ERNA signature.

The multi-modal findings presented in this paper suggest that ERNA is produced by projection-specific synaptic effects of indirect pathway fiber activations; namely, the reciprocal connectivity of the GPe-STN circuit. Further, ERNA appears to be representative of spatially defined neuronal circuit signature that is linked to the therapeutic potential of subthalamic DBS. This insight may be instrumental for the design of future adaptive DBS algorithms that build on constant monitoring and selective manipulation of the engaged subcortical circuit to carefully tailor the neurocircuit intervention via DBS.

# Methods

## Patients
Recordings were obtained from 30 patients with idiopathic PD during awake DBS surgeries (details provided in Supplementary Table 1). All experiments conformed to the guidelines set by the Tri-Council Policy on Ethical Conduct for Research Involving Humans and were approved by the University Health Network Research Ethics Board and each patient provided written informed consent.

## Microelectrode recordings and stimulation
Intracranial microelectrode recordings were acquired during awake DBS surgeries (at least 12 h since the last dose of antiparkinsonian medication) using two closely-spaced microelectrodes (15–25 μm tips; ~600 um spacing; Fig. 1)[9,35]. Recordings were obtained at ≥10 kHz sampling frequency using two Guideline System GS3000 amplifiers (Axon Instruments, Union City, USA) and digitized using a CED1401 data acquisition system with Spike2 v7 software (Cambridge Electronic Design, Cambridge, UK). Microstimulation was delivered using a constant-current stimulator (Neuro-Amp1A, Axon Instruments, Union City, USA). All stimulation within this study was applied with 0.3 ms biphasic pulses with variable stimulation intensity, frequency, and train duration, according to the individual experiment.

Neuronal firing and evoked field potentials were recorded with one microelectrode during stimulation delivered from the adjacent microelectrode at the same depth. Characterization of ERNA (e.g., Fig. 2A) involved measurements of the first (P1) and second (P2) peaks after each stimulation pulse.

During intraoperative mapping procedures, the dorsal border of the STN was identified by increased background noise and irregular neuronal firing at ~20–60 Hz. Neurons within the substantia nigra pars reticulata (SNr) neurons had higher firing rates and more regular firing patterns[50].

## Relationship between the ERNA waveform and STN spiking
To analyse the relationship between ERNA waveform peaks and neuronal firing, stimulation trains at 100 Hz were applied at 30, 50, and 100 μA stimulation intensities for 10 s each at a total of 20 recording locations with well-isolated single-neuron activity (Fig. 2). ERNA waveforms were obtained by averaging interstimulus waveforms at each stimulation intensity. For group-level averages (Fig. 2B), waveforms were first normalized with respect to the average pre-stimulation root mean squared (RMS) amplitude of the local field potential (10–250 Hz). At each recording site, single neuron activity was also recorded, and group-level peristimulus spiking histograms were generated (Fig. 2C).

To investigate relationships between the amplitude of the interstimulus ERNA waveform and neuronal spiking, the log of the rectified ERNA amplitude was plotted against the averaged interstimulus neuronal firing (Fig. 2D); both constrained to the 4–9 ms time window that captures P1 and P2 of the ERNA waveform (see Fig. 2B for reference). To investigate the patterned nature of the relationship between the interstimulus waveform and neuronal firing, log of the ERNA P2 amplitude was plotted against the local minimum of neuronal firing in the 5–7 ms time window (Fig. 2E). For statistical analysis of across-neuron effects, generalized linear mixed-effects regression models

were applied (log ERNA amplitude as the predictor variable and interstimulus average firing as the response variable). To probe the consistency of the polarity of correlations between the ERNA waveform and interstimulus spiking at the within-neuron level, the log ERNA amplitude was correlated with changes in interstimulus spiking across the three stimulation intensities recorded at each site using nonparametric Spearman correlations due to low numbers of observations ($n = 3$) per site. Then, Rho-values were compared to 0 using a 1-sample, 2-tailed $t$ test. All $p$ values were adjusted for multiple comparisons using the Bonferroni correction (4 hypotheses).

## Temporal dynamics of the ERNA waveform

Temporal dynamics of ERNA were studied at the beginning of stimulation trains at 100 Hz stimulation frequency (Fig. 3) at a total of 12 recording locations (100 μA stimulation intensity). At these recording locations, stimulation was also applied at 10, 20, 30, and 50 Hz stimulation frequencies to investigate frequency-specificity.

To assess temporal dynamics across successive stimuli at 100 Hz stimulation frequency, the amplitudes of the ERNA P1 (with respect to the immediate pre-stimulus baseline) and P2 (with respect to interstimulus trough) were extracted using a custom interface developed in Python. P1 and P2 amplitudes (normalized with respect to the pre-stimulation RMS amplitude of the local field potential) were then plotted to visualize the temporal dynamics on a group level (Fig. 3B). In a subsequent step, the dynamics of P1 (calculated as the average amplitude of $P1_{40-45th}$ divided by $P1_{first}$) were correlated with the dynamics of P2 ($P2_{40-45th}/P2_{first}$) using Spearman's correlation (Fig. 3C). Temporal dynamics of ERNA waveforms at lower stimulation frequencies were also characterized (available in Supplementary Fig. 1).

## Computational interrogation of ERNA circuit architecture and dynamics

The empirical results inspired the development of a conceptual framework that may explain the synaptic origin of the ERNA waveform (Fig. 3D, E). Specifically, we developed a computational model in which DBS impulses simultaneously activated cortical and GPe inputs to STN, STN outputs projecting to GPe, and GPe-GPe synapses[16,32] by invasion of axon collaterals[17] of GPe-STN projections (Figs. 3D and 4A). Each of these synapses was subject to short-term synaptic dynamics derived from previous physiological works; wherein cortico-STN synapses rapidly depress[10], GPe-STN and GPe-GPe synapses initially depress to ~65% but are resilient thereafter[9,10], and STN efferent synapses are resilient[12,27] (Figs. 3E and 4B). ERNA P1 is conceptualized as being the result of a net inhibition produced by direct activation of inputs to STN, which slowly depresses over successive stimuli. ERNA P2 is conceptualized as feedback inhibition from GPe, which grows with successive stimuli due to a strengthening of the net excitation produced at GPe (due to static excitation coupled with slowly depressing inhibition) (Fig. 3E).

The model constructed was simulated in PyNN[51] using NEST as the simulation environment[52]. The model consisted of two groups of neurons, STN (100) and GPe (100). The number of neurons was chosen to balance computational expense with variability in connectivity between neuronal populations. STN neurons were modeled via a Poisson firing model, at a rate of 39.9 Hz[35]. GPe neurons were modeled via an exponential conductance based integrate and fire (IF_cond_exp) in NEST. The connectivity modeled between regions was done to allow for the approximation of the mean output of the GPe network being simulated. Each GPe neuron modeled received between 18 and 22 synapses from the STN population, distributed uniformly. GPe neurons were further modeled to fire reciprocally upon each other with different distributions, ranging from [1–5] to [20–25], representing low and high reciprocal connectivity respectively[25], based on a review[53] detailing the number of synaptic processes from STN-GPe and GPe-GPe, scaled down in number and weight to fit in the abstract model. Parameters of the GPe neurons were tuned to match the firings

rates observed ON (~100 Hz) and OFF (~40.5 Hz) stimulation based on observation in rhesus monkeys[12].

Modeling DBS impact considered both afferent and efferent activations. Afferent activations were modeled based on a previous work[35], which considered 500 afferent synapses to the STN, 45% of which were excitatory (via cortex) and 55% of which were inhibitory (via GPe). Excitatory inputs were modeled to depress at a rapid rate, as described in rodent slice work[10], while inhibitory inputs also depressed marginally but were generally resilient as previously described in rodent slice[10], patients with PD[9], and in silico[54]. The model also considered activations of GPe-GPe synapses via antidromic invasion of axon collaterals[17] of STN-GPe projections. Finally, efferent activations were modeled such that each of the GPe neurons connected to STN would receive an orthodromic pulse, activating a static STN synapse to simulate resilient downstream synaptic activations[12,27]. A schematic of all synapses activated by DBS is shown in Fig. 4A.

The Tsodyks-Markram (TM) model[55] was used to implement short-term synaptic plasticity at each activated synapse. Its equations are the following:

$$\frac{du}{dt} = \frac{-u}{\tau_{\text{facil}}} U \tag{1}$$

$$\frac{dr}{dt} = \frac{1-r}{\tau_{\text{rec}}} - u^{+} r^{-\delta(t-t_{\text{sp}})} \tag{2}$$

$$\frac{dI}{dt} = \frac{-I}{\tau_{\text{s}}} + A u^{+} r^{-\delta(t-t_{\text{sp}})} \tag{3}$$

In the above, $u$ is the utilization probability representing the likelihood of neurotransmitter release due to a hypothetical influx of calcium ions. The delta function ($\delta$) makes it such that $u$ is increased by a factor of $U(1-u)$ when a spike arrives, otherwise the potential decays with a constant of $\tau_{\text{facil}}$. The variable $r$ behaves in the opposite manner, decreasing on spike arrival, and otherwise approaches a value of 1 over time with a constant of $\tau_{\text{rec}}$. In the TM model, $U$ is representative of the quantal increase of u produced by a presynaptic spike, while A is the synaptic efficacy of the connection. Equation (3) represents the presynaptic current, which decays at a constant $\tau_{\text{s}}$, is specific to the type of synapse, either excitatory or inhibitory, which have values of 3 ms and 5 ms respectively. Visualization of the dynamics of each activated synapse and TM model parameters are shown in Fig. 4B.

As done in previous works[35], an Ornstein–Uhlenbeck (OU) process with a time constant of 5 ms was used to represent the effect of synaptic noise in the GPe population. The equation of the OU process can be written as:

$$\frac{dx}{dt} = \frac{x(t) - \mu}{\tau} + \alpha \sqrt{\frac{2}{\tau}} \xi(t) \tag{4}$$

Where $\xi$ is a random number with a normal distribution with a mean of 0 and a unit standard deviation, $\tau$ being the time constant, while $\mu$ and $\alpha$ represent the mean and standard deviation of $x$.

The neuron model in use from NEST is IF_cond_exp which is described by the following equations:

$$C\frac{dV}{dt} = -g_{\text{L}} I_{\text{leak}} - I_{\text{spike}} - I_{\text{syn}} - I_{\text{e}} \tag{5}$$

$$I_{\text{leak}} = C\frac{V - V_{\text{p}}}{\tau_{\text{p}}} \tag{6}$$

$$I_{\text{spike}} = C\left(\frac{dV}{dt}\right)_{V=V_{\text{th}}}^{-1} (V_{\text{th}} - V_{\text{r}})\delta(V - V_{\text{th}}) \tag{7}$$

Where $I_{syn}$ is the current created from the synapse model detailed above. $\tau_p$ in Eq. (6) is the passive membrane time constant.

To explore the importance of each network feature, we also varied stimulation frequencies, connectivities, and plasticity dynamics. In particular, as the temporal dynamics of ERNA are only observed during HFS, we simulated a range of frequencies from 10 to 100 Hz (Fig. 4C; Supplementary Fig. 1) to corroborate the importance of frequency specificity[25]. We also tested the model without GPe-GPe connections (Fig. 4C) to assess the importance of the synaptic competition at the level of GPe that is hypothesized to give rise to ERNA P2[25]. Lastly, we tested the model with a depressing, non-static STN synapse (Fig. 4C), to corroborate the importance of STN efferent synaptic resiliency[12].

### Structuro-functional topology and clinical relevance of ERNA

To investigate the topographic organization of ERNA across the spatial extent of STN, stimulation was delivered along the surgical trajectory in 0.5 mm increments, while noting the electrophysiologically defined entry and exit points of the STN. For each stimulation train, inter-stimulus waveforms were generated and the peak-to-trough P2 amplitudes were measured. For group-level analysis in electro-physiological space (Fig. 5A, B), the amplitudes were plotted with respect to depth along the trajectory with the STN entry as a reference point.

ERNA amplitudes were also mapped in MNI space (Fig. 5C, D). To do this, postoperative DBS electrode localizations was performed in Lead-DBS[56], as these DBS stereotactic trajectories are identical to those used for intraoperative microelectrode recordings. For each patient, the post-operative scan (MRI or CT) was rigidly co-registered to the pre-operative MRI scan, and DBS electrodes were manually localized by two experienced users (A.V. and J.G.). Then, pre- and postoperative images, along with the electrodes were normalized to MNI space. Subcortical refine transform in Lead-DBS was used to account for brain shift. The point of intersection of the electrode trajectory and atlas-based STN[57] position were determined. The MER Analysis tool from Lead-DBS was used to determine the MNI and patient native space coordinates of the intersection between the reconstructed electrode and the entry point model STN. Then, a vector of the reconstructed microelectrode trajectory was created using the coordinates of the point of entry into the STN and the center coordinates of the contact. The coordinates of each 0.5 mm microelectrode recording location beginning at STN entry were calculated according to the lead trajectory using Euclidian transformation from native into MNI space. FSLmaths was used to create single voxel coordinates for each recording site, along each trajectory; and each point was assigned its corresponding normalized ERNA amplitude value. Finally, this MNI-space point cloud was transformed into a 3D "heatmap" using Gaussian interpolation in Plotly[58] (Fig. 5C; superimposed onto an STN atlas rendering[59]). For visualization purposes, the figure also portrays the location of the derived group-level ERNA hotspot[28]. Clinical improvement with optimized postoperative DBS settings was correlated (Spearman's correlation; p-values adjusted for multiple comparisons using the Bonferroni correction; 2 hypotheses) with the Euclidean distance between the DBS clinical sweet spot and the patient-specific ERNA hotspots (Fig. 5D). Clinical improvements by DBS were measured with the patients withdrawn from their dopaminergic medication and considered both the Movement Disorders Society−Unified Parkinson's disease Rating Scale Part III total motor score and akinetic-rigid subscore in the side of the body contralateral to stimulation. In a final step, the fiber activation profile[48] of the group level ERNA hotspot (1 mm radius) was characterized using the subcortical Petersen fiber atlas (human histological and structural MRI data which has been further refined by expert neuroanatomists)[26], which considered M1 (face, upper extremity, and lower extremity), SMA, and premotor hyperdirect pathways, as well as GPe-STN and GPe-STN indirect pathway fibers (Fig. 5E; fibers are superimposed onto STN and GPe atlas renderings[59] and a 7T brain scan in MNI 152 space[60]). Relative percentages of the sensorimotor hyperdirect and indirect pathway fibers passing through the ERNA hotspot were reported (Fig. 5F).

### Reporting summary

Further information on research design is available in the Nature Portfolio Reporting Summary linked to this article.

## Data availability

Experimental data (Figs. 2, 3, and 5) in this study are provided[61] in the Supplementary Material/Source Data files; sources and access are summarized in Supplementary Table 1. Source data are provided with this paper.

## Code availability

Codes for the ERNA peak extraction interface[62], 3D heatmap generator[63], and computational modeling[64] are provided.

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

## Acknowledgements

This project has been made possible with the financial support of Health Canada, through the Canada Brain Research Fund, an innovative partnership between the Government of Canada (through Health Canada) and Brain Canada, and of the Azrieli Foundation (L.M.); Natural Sciences and Engineering Council (NSERC) RGPIN-2022-05181 (L.M.); Canadian Institute for Health Research (CIHR) PJT 191880 (L.M., S.K.K., A.F.); Deutsche Forschungsgemeinschaft (DFG, German Research Foundation) 424778381 - TRR 295 (L.A.S., A.A.K.); Hertie Foundation P1230041 (L.A.S.). The authors would like to thank the participants for their invaluable contributions to the work.

## Author contributions

L.A.S. wrote the first version of the manuscript; L.A.S. performed experimental data analyses and generated figures. S.S. supported experimental data analyses; D.C. developed computational models in consultation with M.L.; J.G. and A.V. performed neuroimaging analyses in consultation with A.B.; M.S. developed the 3D hotspot map; M.R.P., W.D.H., and A.K. discussed the results and research direction; M.J. and A.F. acquired and provided clinical information; M.H., S.K.K., and A.M.L. performed surgeries and enabled intraoperative data collection; all authors read and revised the manuscript; L.M. supervised research.

## Competing interests

L.M., M.R.P., M.L., and W.D.H. hold intellectual property related to predicting and modeling neuronal responses to DBS (patent publication number: 20220152396; application number: 17/527,042). All other authors declare no competing interests related to this work.
