## [Peer Review File · Nature Communications]

Neural signatures of indirect pathway activity during subthalamic stimulation in Parkinson's diseaseREVIEWER COMMENTS

Reviewer #1 (Remarks to the Author):

Overview

This is a very interesting and very original paper that seeks to elucidate the synaptic and axonal mechanisms of an important and recently discovered phenomenon, Evoked Resonant Neural Activity (ERNA). The authors are uniquely positioned to do this study, leveraging their intraoperative recording set up that is customized to provide independent control of two closely spaced microelectrodes. This allows recording of both LFP like phenomena, and single units, in close proximity to another electrode that provides the stimulus. The computational and imaging components add to the paper but the physiology is the most novel element. There are a lot of great data here and the fact that the authors took a “deep dive” on understanding the basis for ERNA is much appreciated. There are several general issues that reduce the value of the paper in its current form, and a number of specific, more minor ones.

Major issues

1) At many points in the paper, especially discussion, the language is complex, with long sentences whose exact meaning is elusive (example below). The authors need to rewrite, these parts to more clearly elucidate their key points in a more accessible way with cleaner, simpler language.

2) A major point of the paper is that ERNA is a property of the indirect intrinsic basal ganglia pathway. But, in both 3D and 4A, the stimulation of hyperdirect inputs to STN places a role in the model and thus would seem to play a role in generating ERNA. This seems to contradict the point that ERNA is a property of the indirect pathway (which classically is striatal D2 to GPe to STN to GPi). Could ERNA still happen without hyperdirect inputs? If so why does the hyperdirect input appear to play a role in the author's model of ERNA?

3) The authors make the important point that ERNA can also be observed during pallidal DBS, not just subthalamic. The introduction and discussion indicate that involvement of GPe is likely a common mechanism of both. Could they speculate more on the synaptic mechanism of pallidal stim? Is it that the therapeutic target of pallidal stim is necessarily GPe (eg the therapeutic contact should be in or at the border of GPe) or is retrograde activation of GPe-GPi fibers more likely to engage the relevant GPe synapses? Can they account for fact that pallidal ERNA is lower amplitude than subthalamic ERNA?

Comments on specific sections

Intro: “Parkinson’s disease (PD) is a common movement disorder that occurs after degeneration of nigrostriatal dopaminergic projections..” this isn’t quite accurate... the motor signs occur after nigrostriatal degeneration, but the disorder starts much earlier than that.

“An integrative account of these synaptic circuit activations and their importance for neuronal circuit engagement in the subcortex is missing; which motivated the contextualization of these mechanistic phenomena within this work” This should be said in a simpler way

Results/figures

The authors emphasize that stimulation results in “patterned” inhibition of STN neuronal firing. The specific pattern generated should be stated. Is that suppression of STN firing at a particular time point after a stimulation pulse, but not at other time points, is critical for the emergence of ERNA? If so it needs to be explained more clearly as this patterning received a lot of emphasis throughout the paper.

Figure 3D – this schematic seems like it would be very useful for understanding the mental model of how ERNA comes about from synaptic interactions. I tried to understand it but failed. The three parts I, ii, and iii are probably sequential time points but this isn’t specified. In the legend, it wasn’t clear to me what comments referred to 3D versus 3E. the exact meaning of the letters a and b in the figure aren’t clear to me. The schematic needs to be simplified and/or explained much better.

5c – the orientation of the STN is confusing. The “dorsolateral” arrow suggests that this is a coronal view (since the coronal plane contains a dorso-ventral axis and medial-lateral axis). But the STN is not oriented the way one would expect in the coronal plane (would be obliquely).

5f- not mentioned in the legend. The results do describe it: “Fibers crossing through the group level ERNA hotspot (1 mm radius) consisted of 30% GPe-STN, 30% STN-GPe, 12% SMA-STN, 10% premotor-STN, and 19% M1-STN fibers”. From results and legend to fig 5, its not clear where these percentages come from. The methods refers to a ‘fiber atlas” but very little information is given to understand how to interpret 5f and at minimum the derivation of these from a specific fiber atlas should be mentioned when giving those results and some statement of what that atlas means (was it from human histology, for example)

Discussion:

The authors emphasize the point that ERNA is linked to the therapeutic outcome of DBS, and indicate that modulation of the indirect pathway is important in the therapeutic mechanism, but don’t really explain the exact link between ERNA and therapeutic benefit. Does ERNA itself produce a downstream effect that is responsible for the efficacy of STN DBS? Or is it just a marker for engagement of a population of GPe cells that themselves are important for the therapeutic mechanism? Further, could ERNA produce the beta suppression that is a known hallmark of therapeutic DBS? The answers of course require speculation but is worth doing here given the very detailed dissection of basal ganglia microcircuitry.

A recent paper showed that ERNA occurs in dystonia, which seems to suggest it is related to the generic basal ganglia architecture rather than to specific parkinsonian derangements (such as increased beta band activity). Could the presence of ERNA in dystonia indicate that therapeutic DBS in that condition must also engage similar synaptic mechanisms in GPe? This is worth a comment in the discussion.

There are many sentences that are too long, and are hard to follow, some examples below:

“The fact that ERNA can be monitored using the macrocontacts of DBS electrodes during clinically-relevant stimulation highlights the translational relevance of the underlying synaptic / neuronal circuit signature as suggested by the findings presented in this study” – not sure what this sentence means

“Our finding that patterned neuronal inhibition is temporally locked to peaks of the ERNA waveform provides a high spatiotemporal resolution neurophysiological readout that not only implicates the subcortical orchestration of ERNA, but also substantiates the hypothesis that this electrophysiological signature is a representation of GPe-mediated inhibition” – this sentence is very hard to follow

“While convergence of indirect vs hyperdirect pathway fibers at the level of STN has been shown to be instrumental in the orchestration of pathological subcortical oscillatory activity and neuronal synchrony, there has been debate about the level of discretization of respective fiber activations during subthalamic DBS” – by “level of discretization” do you mean whether all pathways are active or just some? The sentence is unclear.

Reviewer #2 (Remarks to the Author):

Steiner and colleagues explored micro and macro scale origins of ERNA- an important physiological marker of deep brain stimulation efficacy when applied to the STN. Cellular origins of this physiological marker still remains unknown and could potentially inform DBS targeting and lead to development of new stimulation approaches. This is a well written study which presents a novel hypothesis and would be suitable for publication after major revisions.

My only feedback to the authors is regarding their computational model which is an important element of this study tying micro electrode recordings to mapping in normative anatomical space.

(1) Authors used an integrate and fire model to represent 100 STN and 100 GPE neurons. In this particular context, when their main hypothesis is the engagement of the indirect pathway (which would hyperpolarize STN neurons), I would have expected the authors to utilise a STN model that can elicit rebound excitation after hyperpolarization (Bevan et al 2000, Terman et al 2002). The authors should either update their theoretical approach or convincingly motivate their modelling choices (i.e. why in this experimental context, we should not be expecting rebound activation).

(2) In the absence of the computational model exhibiting appropriate membrane properties, it becomes difficult to be convinced by the following statement "In particular, the initial peak is conceptualized to be the result of sustained GPe-mediated inhibition to STN, whereas the second peak is the result of synaptic competition at the level of GPE which feeds back to STN to produce recurrent inhibition." which entirely relies on the computational model.

NCOMMS-23-25757 [R1]

(Neural signatures of indirect pathway activity during subthalamic stimulation in Parkinson's disease)

The authors would like to thank the Editorial team and Reviewers for providing us the opportunity to improve the manuscript content and data presentation through insightful feedback. Point by point responses are found below.

Responses to Reviewer #1

Overview:

This is a very interesting and very original paper that seeks to elucidate the synaptic and axonal mechanisms of an important and recently discovered phenomenon, Evoked Resonant Neural Activity (ERNA). The authors are uniquely positioned to do this study, leveraging their intraoperative recording set up that is customized to provide independent control of two closely spaced microelectrodes. This allows recording of both LFP like phenomena, and single units, in close proximity to another electrode that provides the stimulus. The computational and imaging components add to the paper but the physiology is the most novel element. There are a lot of great data here and the fact that the authors took a “deep dive” on understanding the basis for ERNA is much appreciated. There are several general issues that reduce the value of the paper in its current form, and a number of specific, more minor ones.

Thank you for your kind appraisal and thoughtful comments, which helped us to improve the quality of our manuscript and data presentation.

Major:

1) At many points in the paper, especially discussion, the language is complex, with long sentences whose exact meaning is elusive (example below). The authors need to rewrite, these parts to more clearly elucidate their key points in a more accessible way with cleaner, simpler language.

Thank you for this important feedback.

Amendments to the manuscript: We have rewritten various sentences outlined by the reviewer (see below; in responses to “comments on specific sections”). We have moreover edited the entire manuscript to convey key messages in clearer and simpler language. These general edits throughout can also be identified by blue text within the manuscript.

2) A major point of the paper is that ERNA is a property of the indirect intrinsic basal ganglia pathway. But, in both 3D and 4A, the stimulation of hyperdirect inputs to STN places a role in the model and thus would seem to play a role in generating ERNA. This seems to contradict the point that that ERNA is a property of the indirect pathway (which classically is striatal D2 to GPe to STN to GPi). Could ERNA still happen without hyperdirect inputs? If so why does the hyperdirect input appear to play a role in the author's model of ERNA?

The reviewer raises an important point regarding the relevance of hyperdirect pathway inputs for the generation of ERNA. Hyperdirect pathway inputs to STN have been incorporated into our model to provide an anatomically accurate representation of the circuit architecture. Because activation of cortical afferents to the STN is a phenomenon that occurs during STN-DBS, these fibers were not omitted from the model. However, an important detail which is supported by empirical data in STN slices (Steiner et al., *J Neurosci*, 2019) and is incorporated into the model is that cortical inputs to the STN rapidly and robustly depress during high-frequency stimulation (outlined in Fig. 4B; please note that other synaptic currents are able to maintain neurotransmitter release during high frequency activations, as per implicated references). Thus, despite the physical presence of cortical axonal inputs, the functional contribution to the produced synaptic current in STN is effectively negligible after the first few impulses due to inherent functional properties of these synapses (rapid depression).

Inspired by the Reviewer’s comment, we have used our model to probe the relevance of hyperdirect inputs in more detail (and conceptualize the aforementioned points). To this end, we have implemented a new *in silico* condition in which hyperdirect inputs are omitted. In this condition, ERNA peaks show similar dynamics as compared to rapidly depressing hyperdirect inputs. This new implementation demonstrates that hyperdirect pathway *depression* during high-frequency stimulation has a similar effect as outright hyperdirect pathway omission; therefore, reinforcing the point that hyperdirect pathway inputs are of negligible importance towards the generation of the ERNA waveform.

Amendments to the manuscript:

The following text has been added to the Discussion (Pg. 14/15):

“Because of the rapid depression of hyperdirect inputs during HFS, the effect of hyperdirect pathway stimulation in the generation of ERNA can be considered negligible. This is corroborated by the *in silico* findings presented in this study, which show very similar dynamics of ERNA in the absence of hyperdirect inputs as compared to rapidly depressing hyperdirect inputs (Supplementary Fig. 3)”.

We have also added this implementation to the Supplementary Material:

Supplementary Figure 3 – Modeling results in the presence and absence of rapidly depressing hyperdirect inputs to STN. (Upper panels) STN conductances in response to HFS. (Lower panels) In silico dynamics of first (dark blue) and second peak (light blue) of the ERNA waveform. Note that ERNA peaks

show similar dynamics in the (i) presence and (ii) absence of rapidly depressing hyperdirect inputs to STN. This suggests that hyperdirect pathway inputs are likely of negligible importance towards the generation of the ERNA waveform.

3) The authors make the important point that ERNA can also be observed during pallidal DBS, not just subthalamic. The introduction and discussion indicate that involvement of GPe is likely a common mechanism of both. Could they speculate more on the synaptic mechanism of pallidal stim? *[for simplicity, we have broken this comment up into three parts, retaining black text for Reviewer questions and blue text for answers]*

In a recent scientific commentary (Steiner & Milosevic. *Brain Comms*, 2023) and review article (Neumann, Steiner, Milosevic. *Brain*, 2023) we proposed that, similar to STN ERNA, the initial peak of GPi ERNA is a result of direct activation of afferent inputs (producing a net hyperpolarization, due to the greater abundance of GPe inputs compared to STN). Concurrent to the activation of afferent inputs, we proposed that GPi-DBS may additionally (through invasion of axon collaterals and fibers of passage) invade the same circuit that is responsible for the production of STN ERNA (i.e., reciprocal connections between STN and GPe). Through antidromic activation of GPe and STN afferents, and eventual invasion of axon collaterals, GPi DBS can produce neurotransmitter release at remote sites (i.e., STN-mediated release of glutamate in GPe, and GPe-mediated release of GABA in STN). The excitation of GPe would thereafter lead to recurrent inhibition of GPi (and STN), and therefore, the second peak of the ERNA waveform. For reference, below is a figure (and figure legend) from Neumann, Steiner, Milosevic. *Brain*, 2023 conceptualizing this common network activation phenomenon:

Neumann, Steiner, Milosevic (Brain, 2023) – Figure 3 _ Mesoscale effects of DBS. (ii) Subthalamic nucleus (STN) and (iii) globus pallidus internus (GPi) during 100 Hz stimulation, as well as hypothesized circuit activation profiles that would explain the emergence of ERNA (adapted from Steiner et al. 2023, Brain Communications). In STN, each stimulus would produce a net inhibitory response in STN, as well as concurrent excitation of globus pallidus externus (GPe), resulting in feedback inhibition in STN. The same is hypothesized for GPi ERNA via invasion/activation of collateral projections and axons of passage of the reciprocal STN-GPe connectivity. Thus, each of the ERNA waveform peaks is likely a substrate of inhibitory input via GPe. An important note is that the spike firing patterns in STN are only achieved when using subthreshold stimulation amplitudes, which do not cause complete suppression of neuronal firing. High-frequency stimulation (HFS) at clinically relevant intensities would result in the complete suppression of firing and the elicitation of large amplitude ERNA waveforms. In GPi, patterned firing seems to manifest when spike firing re-emerges (...) after depression of striatal GABAergic inputs, likely unmasking inhibitory-excitatory GPe-STN competition (...).

Additional support for the suggested mesocircuit activation comes from observations that: (1) effective GPi-DBS produces inhibition of neuronal firing in STN (hypothesized to be

mediated by GPe; Liu et al., *J Neurophys*, 2012); and (2) STN-DBS produces ERNA in GPi (Schmidt et al., *Brain Stim*, 2020).

While the main focus of the present work is on mechanistic aspects of STN ERNA, we have nevertheless added further speculative details about GPi ERNA per the request of the Reviewer.

Amendments to the manuscript:

The following text has been added to the Discussion (Pg. 15):

“As such, GPi DBS may in fact also invade the reciprocal STN-GPe mesocircuit network which underlies the generation of ERNA. In GPi, the initial ERNA peak is likely a result of direct activation of afferent inputs (producing a net hyperpolarization, due to the greater abundance of GPe inputs compared to STN). Through concurrent antidromic activation of GPe and STN afferents, and invasion of axon collaterals, GPi-DBS can produce STN-mediated release of glutamate in GPe, and GPe-mediated release of GABA in STN. The excitation of GPe would thereafter lead to recurrent inhibition of GPi, and therefore, the second peak of the ERNA waveform.”

Is it that the therapeutic target of pallidal stim is necessarily GPe (eg the therapeutic contact should be in or at the border of GPe) or is retrograde activation of GPe-GPi fibers more likely to engage the relevant GPe synapses?

Thorough scrutinization of synaptic activation responses in GPi (e.g., ERNA & striatal-mediated responses, etc.) is presently a topic of ongoing investigation in the lab, in which we plan to relate evoked responses to stimulation location (to be able to answer the Reviewer’s question directly in subsequent work). At present however, we are happy to speculate. As is outlined in the above amendment, we suspect that it is the result of retrograde activation of fibers from GPe-to-GPi (which have collaterals to STN) and STN-to-GPi (which have collaterals to GPe). Optogenetics works have indeed suggested that global activation of GPe does not produce therapeutic benefit in parkinsonian rodents (Mastro et al., *Nat Neurosci*, 2017), whereas cell-type specific GPe interventions do, which (speculatively) may be achieved by DBS through retrograde activation of GPe-GPi fibers.

Amendments to the manuscript:

As above (Pg. 15), in addition to the following clarification (Pg. 15):

“Moreover, subsequent optogenetic works have also shown that direct somatic inhibition of STN also produced antiparkinsonian benefits,⁴⁵ and that activation of populations of GPe neurons that selectively project to STN could also produce long-lasting antiparkinsonian benefits (whereas global activation of GPe did not).^{46”}

Can they account for fact that pallidal ERNA is lower amplitude than subthalamic ERNA?

This is indeed a very interesting question. At present, we can only speculate about these relative differences. We hypothesize that differences in ERNA amplitudes may be a reflection of the relative innervation strengths from GPe. STN receives the vast majority of its GABAergic input from GPe (Bevan et al., *Prog Brain Res*, 2007) while the GABAergic input to GPi is more mixed, containing inhibitory inputs from striatum, GPe, and mutual synaptic connectivity. Greater relative innervation of GPe-to-STN as compared to GPe-to-GPi may therefore be the reason for greater ERNA amplitudes in STN.

Amendments to the manuscript:

The following text has been added to the Discussion (Pg. 15):

“A greater relative innervation of inhibitory inputs from GPe-to-STN as compared to GPe-to-GPi (which also receives inhibitory inputs from other sources; namely, striatum) may explain observations of greater ERNA amplitudes in STN compared to GPi.”

Comments on specific sections:

Intro:

“Parkinson’s disease (PD) is a common movement disorder that occurs after degeneration of nigrostriatal dopaminergic projections..” this isn’t quite accurate... the motor signs occur after nigrostriatal degeneration, but the disorder starts much earlier than that.

Thank you for the attentive comment.

Amendment to the manuscript:

The following clarification has been made (Pg. 3):

“Parkinson’s disease (PD) is a common movement disorder that ~~occurs after~~ is associated with progressive degeneration of nigrostriatal dopaminergic projections.”

“An integrative account of these synaptic circuit activations and their importance for neuronal circuit engagement in the subcortex is missing; which motivated the contextualization of these mechanistic phenomena within this work” This should be said in a simpler way

Thank you for the attentive comment.

Amendment to the manuscript:

The following clarification has been made (Pg. 3):

“An integrative account of these synaptic subcortical circuit activations ~~and their importance for neuronal circuit engagement in the subeortex~~ is missing, which motivated the ~~contextualization of these mechanistic phenomena within this work~~ presented in this study.”

Results/figures:

The authors emphasize that stimulation results in “patterned” inhibition of STN neuronal firing. The specific pattern generated should be stated.

Thank you for the attentive comment. We now explicitly specify the pattern of inhibition of STN neuronal firing.

Amendment to the manuscript:

The following clarification has been made to the Results (Pg. 10):

“These data demonstrate a relationship between the shape of the ERNA waveform and the similarly patterned neuronal suppression in the interstimulus interval (inhibition of neuronal activity that is time locked to peaks of the ERNA waveform, whereby the strength of the inhibition is proportional to size of the respective peak).”

Is that suppression of STN firing at a particular time point after a stimulation pulse, but not at other time points, is critical for the emergence of ERNA? If so it needs to be explained more clearly as this patterning received a lot of emphasis throughout the paper.

The data presented in this paper suggests that the ERNA waveform is a readout of the recurrent synaptic input activation of indirect pathway fibers. STN neuronal suppression during HFS is a consequence of GPe-mediated inhibitory activations (i.e., ERNA) and as such, each peak of the ERNA waveform *produces* inhibition of neuronal firing (Fig 2 of manuscript). STN firing itself, while being patterned by ERNA (i.e., subject to recurrent inhibition), is not essential for the emergence of ERNA. In fact, when ERNA (inhibitory signature) amplitude is large, neuronal firing is more suppressed; thus, large amplitude ERNA waveforms impose potent inhibition, and therefore exist in the complete absence of spiking. This is supported by data displayed in Figure 2 that suggests that stronger neuronal suppression is correlated with higher amplitudes of ERNA. Findings of greater ERNA amplitudes being related to better therapeutic results (e.g., Sinclair et al., *Ann Neurol*, 2019) likely suggest stronger entrainment of GPe-mediated inhibition.

Amendment to the manuscript:

The following clarification has been made in the Discussion (Pg. 13):

~~“Our finding that the ERNA waveform peaks are temporally locked to inhibitions of STN single-neuron activity that patterned neuronal inhibition is temporally locked to peaks of the ERNA waveform provides a high spatiotemporal resolution neurophysiological readout that not only implicates the subcortical orchestration of ERNA, but also substantiates the hypothesis that ERNA this electrophysiological signature is likely a GPe-mediated signature (given that GPe is the major source of inhibition to STN).”~~

Figure 3D – this schematic seems like it would be very useful for understanding the mental model of how ERNA comes about from synaptic interactions. I tried to understand it but failed. The three parts i, ii, and iii are probably sequential time points but this isn't specified. In the legend, it wasn't clear to me what comments referred to 3D versus 3E. the exact meaning of the letters a and b in the figure aren't clear to me. The schematic needs to be simplified and/or explained much better.

Thank you for the opportunity to clarify. 1a and 1b happen simultaneously, hence use of the same number. 1a: Recruitment of inputs to STN (cortical and GPe; the net effect is sustained inhibition in STN, i.e., ERNA P1); 1b: Orthodromic recruitment of excitatory STN-GPe efferents & antidromic recruitment of inhibitory GPe-STN afferents with invasion of GPe-GPe axon collaterals (producing synaptic competition at GPe, giving rise to net excitation of increasing strength). 2: This recurrent inhibition in STN is the result of an emerging net excitation in GPe (i.e., 1b), which feeds forward to STN (producing ERNA P2).

Amendments to the figure:

- amended all subheadings for clarity (provided explicit explanations wherever possible)
- specified that 1a and 1b occur simultaneously
- colour-coded the two distinct timepoints represented within each interstimulus interval
- amended figure legend text

... (D) Conceptual schematic to illustrate how STN stimulation may trigger a cascade of synaptic events that may ultimately give rise to the resonant ERNA peak. Left: Schematic representation of activated fibers at two timepoints during a single interstimulus interval (1a/b & 2). Note that (1a) and (1b) are expected to occur simultaneously within a “monosynaptic” time course in response to individual stimuli, whereas (2; blue shading) occurs subsequently/consequently, at a “disynaptic” time course. Right: These synaptic responses change in amplitude across successive interstimulus intervals as a result of short-term synaptic plasticity. Vertical lines represent individual stimuli at HFS. Blue positive-going potentials in STN correspond to direct inhibitory afferent activations (1a-blue), whereas negative-going red potentials represent activations of excitatory cortical inputs (1a-red). 1a: sustained GPe transmission paired with rapidly depressing cortical transmission leads to a sustained net inhibition in STN (i.e., P1 of STN ERNA, filled grey). 1b: sustained GPe transmission paired with even more sustained STN transmission leads to an increasing net excitation in GPe. 2: the increasing net excitation in GPe produces recurrent inhibition of STN (i.e., P2 of STN ERNA, filled black).

5c – the orientation of the STN is confusing. The “dorsolateral” arrow suggests that this is a coronal view (since the coronal plane contains a dorso-ventral axis and medial-lateral axis). But the STN is not oriented the way one would expect in the coronal plane (would be obliquely).

Thank you for this feedback. We agree that the previously used 2D arrow was insufficient to facilitate understanding of the orientation.

Amendment: We have incorporated 3D orientation arrows.

Groupdata: ERNA in normative three dimensional space (MNI)

(Figure 5C) Group data (n = 19; postoperative neuroimaging was unavailable for one patient) of the ERNA amplitude heatmap in MNI space. orientation arrows - d: dorsal; m: medial; p: posterior

5f- not mentioned in the legend. The results do describe it: “Fibers crossing through the group level ERNA hotspot (1 mm radius) consisted of 30% GPe-STN, 30% STN-GPe, 12% SMA-STN, 10% premotor-STN, and 19% M1-STN fibers”. From results and legend to fig 5, its not clear where these percentages come from. The methods refers to a ‘fiber atlas’ but very little information is given to understand how to interpret 5f and at minimum the derivation of these from a specific fiber atlas should be mentioned when giving those results and some statement of what that atlas means (was it from human histology, for example)

Thank you for the attentive comment. We implemented the Petersen atlas that draws from human histological and structural MRI data and has been further refined by expert neuroanatomists (Petersen et al, *Neuron*, 2019). In 5f, the relative percentages of sensorimotor hyperdirect and indirect pathway fibers passing through the ERNA hotspot were reported as a percentage of the total number of fibers passing through the ERNA hotspot.

Amendment to the manuscript:

The following clarification has been made in the Methods (Pg. 9):

“In a final step, the fiber activation profile³⁸ of the group level ERNA hotspot (1 mm radius) was characterized using the subcortical Petersen fiber atlas (human histological and structural MRI data which has been further refined by expert neuroanatomists),²⁵ which considered M1 (face, upper extremity, and lower extremity), SMA, and premotor hyperdirect pathways, as well as GPe-STN and GPe-STN indirect pathway fibers (Fig. 5E). Relative percentages of the sensorimotor hyperdirect and indirect pathway fibers passing through the ERNA hotspot were reported (Fig. 5F).”

Discussion:

The authors emphasize the point that ERNA is linked to the therapeutic outcome of DBS, and indicate that modulation of the indirect pathway is important in the therapeutic mechanism, but don’t really explain the exact link between ERNA and therapeutic benefit. Does ERNA itself produce a downstream effect that is responsible for the efficacy of STN DBS? [*for simplicity, we have broken this comment up into three parts, retaining white text for Reviewer questions and blue text for answers*]

Thank you for raising these relevant questions. Our work suggests that at the level of STN, ERNA likely represents recurrent synaptic inhibition via GPe, as shown in Figure 2. While the amount of inhibition of neuronal firing has been previously shown to be associated with the therapeutic threshold of STN-DBS (Milosevic et al. *JNNP*, 2019), the more widespread cortico-basal-ganglia network effects are yet to be evaluated in detail. Recent work from the group of Andreas Horn has suggested that an overlap exists in the cortical functional (fMRI-based) connectomic profiles associated with STN- and GPi-DBS. We hypothesize (Steiner & Milosevic, *Brain Comms*, 2023) that this high-latency functional overlap in cortex is a downstream / feedforward phenomenon driven by GPe-mediated inhibition throughout the cortico-basal ganglia network, since (i) ERNA is common signature to both STN- and GPi-DBS (Johnson et al., *Brain Comms*, 2023), whereas (ii) fast-latency (antidromic-driven) cortical activations do not occur during effective GPi-DBS (Johnson et al., *J Neurosci*, 2020).

Amendment to the manuscript:

The following clarification has been made in the Discussion (Pg. 15):

“It is perhaps this resonant subcortical mesocircuit phenomenon, which promotes GPe-mediated ~~resonant~~ downstream inhibition throughout the broader basal ganglia network, that also underlies the slower timecourse anticorrelation observed within the convergent functional connectomic profile associated with both of these interventions.⁴⁷”

Or is it just a marker for engagement of a population of GPe cells that themselves are important for the therapeutic mechanism?

Importantly only cell type specific, but not global stimulation of GPe produces therapeutic benefit, which has recently confirmed by preclinical studies. In the 6-OHDA model of PD, optogenetic GPe modulation has been shown to produce therapeutic benefit in a cell-type specific manner (when increasing activity of PV+, but not Lhx6 neurons; Mastro et al, *Nat Neurosci*, 2017). Interestingly, PV+-neurons have been shown to preferentially target STN (Mastro et al, *J Neurosci*, 2015). Thus, stimulation in areas of STN that receive projections from these cells may produce similar circuit activation responses. These areas may be able to be identified by eliciting large amplitudes of ERNA.

Amendment to the manuscript:

The following clarification has been made in the Discussion (Pg. 16):

“Moreover, targeting of the subcortical network that gives rise to ERNA may enable further optimization of STN DBS for therapeutic purposes. Given that ~~It has been shown that~~ the activation of GPe neuronal populations that preferentially project to STN can lead to long-lasting therapeutic benefits that persist beyond stimulation cessation,⁴⁶ our findings may provide a means of selectively targeting these therapeutically-relevant GPe fibers.”

Further, could ERNA produce the beta suppression that is a known hallmark of therapeutic DBS? The answers of course require speculation but is worth doing here given the very detailed dissection of basal ganglia microcircuitry.

Indeed, we hypothesize that GPe-mediated inhibition (which is our proposed electrophysiological basis for ERNA) likely also contributes to the suppression of beta oscillations. Beta oscillations are thought to critically depend on pallidal inputs to STN that are in antiphase to cortical inputs (Baufreton et al., *J Neurosci*, 2005; Cagnan et al., *Brain*, 2015); whereas our understanding of DBS is that it results in the depression of cortical inputs and entrainment of GPe inputs to STN. Thus, these pathway modulations likely cumulatively contribute to the disruption and suppression of circuit phenomena necessary to produce pathological beta oscillations (Steiner et al., *J Neurosci*, 2019).

Amendment to the manuscript:

The following clarification has been made in the Discussion (Pg. 16):

“These observations collectively suggest that subthalamic DBS likely engages the same subcortical mesocircuit network that gives rise to beta oscillations, imposing stimulation-induced suppression of pathological activity through high frequency driving of GPe-mediated inhibition (i.e., ERNA) in tandem with suppression of cortical influence on STN but has the capability to exploit the very same circuit motif to produce a therapeutic neurocircuit intervention that is reflected in ERNA.”

A recent paper showed that ERNA occurs in dystonia, which seems to suggest it is related to the generic basal ganglia architecture rather than to specific parkinsonian derangements (such as increased beta band activity). Could the presence of ERNA in dystonia indicate

that therapeutic DBS in that condition must also engage similar synaptic mechanisms in GPe? This is worth a comment in the discussion.

Indeed, Wiest et al showed that ERNA can be elicited by STN-DBS in dystonia (Wiest et al., *Mov Disord*, 2023). We agree with the reviewer that this may be evidence that ERNA is reflective of a synapse-specific circuit-phenomenon rather than being disease specific.

Amendment to the manuscript:

The following text has been added to the Discussion (Pg. 15):

“Interestingly, ERNA can also be elicited by STN-DBS in dystonia, substantiating that ERNA may be reflective of a synapse-specific circuit-phenomenon rather than being disease specific⁵⁶.”

There are many sentences that are too long, and are hard to follow, some examples below:

“The fact that ERNA can be monitored using the macrocontacts of DBS electrodes during clinically-relevant stimulation highlights the translational relevance of the underlying synaptic / neuronal circuit signature as suggested by the findings presented in this study” – not sure what this sentence means

The sentence was initially intended to highlight usability of ERNA as a biomarker of circuit activation. We agree with the reviewer that this sentence is not necessary to motivate the present study.

Amendment to the manuscript: In order to provide a more coherent discussion, we have deleted the respective paragraph from the Discussion (Pg. 13).

“Our finding that patterned neuronal inhibition is temporally locked to peaks of the ERNA waveform provides a high spatiotemporal resolution neurophysiological readout that not only implicates the subcortical orchestration of ERNA, but also substantiates the hypothesis that this electrophysiological signature is a representation of GPe-mediated inhibition” – this sentence is very hard to follow

We agree with the reviewer that this sentence is unnecessarily complicated.

Amendment to the manuscript:

The following clarification has been made (Pg. 13):

“Our finding that the ERNA waveform peaks are temporally-locked / give rise to inhibitions of single-neuron activity a high spatiotemporal resolution neurophysiological readout that not only implicates the subcortical orchestration of ERNA, but also substantiates the hypothesis that ERNA this electrophysiological signature is likely a GPe-mediated signature (given that GPe is the major source of inhibition to STN).”

“While convergence of indirect vs hyperdirect pathway fibers at the level of STN has been shown to be instrumental in the orchestration of pathological subcortical oscillatory activity and neuronal synchrony, there has been debate about the level of discretization of respective fiber activations during subthalamic DBS” – by “level of discretization” do you mean whether all pathways are active or just some? The sentence is unclear.

Thank you for the opportunity to clarify. With the “level of discretization” we wanted to specify that the relative contributions of engaged pathways, as it relates to therapeutic efficacy, is of ongoing debate (Butenko et al. *Neuroimage Clin.* 2022). However, it is not only important which fibers are activated, but also the functional consequences of such activations (which we believe is the most valuable contribution of our work).

Amendment to the manuscript:

The following clarification has been added to the Discussion (Pg. 14):

“While convergence of indirect vs hyperdirect pathway fibers at the level of STN has been shown to be instrumental in the orchestration of pathological subcortical oscillatory activity^{1,4} and neuronal synchrony,⁴³ there has been debate about the ~~level of discretization of respective fiber activations during~~ relative contributions of engaged pathways to the therapeutic effect of subthalamic DBS.^{6,44} However, beyond these structural fiber profiles, we suggest that projection-specific synaptic dynamics have to be taken into consideration to appreciate the functional consequences of such fiber activations (e.g., cortical suppression paired with GPe entrainment).”

Responses to Reviewer #2:

Steiner and colleagues explored micro and macro scale origins of ERNA- an important physiological marker of deep brain stimulation efficacy when applied to the STN. Cellular origins of this physiological marker still remains unknown and could potentially inform DBS targeting and lead to development of new stimulation approaches. This is a well written study which presents a novel hypothesis and would be suitable for publication after major revisions.

Thank you for your kind appraisal and your comments, which helped us to improve our manuscript and advance our computational modeling.

My only feedback to the authors is regarding their computational model which is an important element of this study tying micro electrode recordings to mapping in normative anatomical space.

(1) Authors used an integrate and fire model to represent 100 STN and 100 GPE neurons. In this particular context, when their main hypothesis is the engagement of the indirect pathway (which would hyperpolarize STN neurons), I would have expected the authors to utilise a STN model that can elicit rebound excitation after hyperpolarization (Bevan et al 2000, Terman et al 2002). The authors should either update their theoretical approach or convincingly motivate their modelling choices (i.e. why in this experimental context, we should not be expecting rebound activation).

Thank you for raising this very important point. We find that conceptually incorporating the rebound burst phenomenon into our framework helps to provide further support of the main message of our findings: that ERNA is a GPe-mediated hyperpolarizing tone that is persistently imposed upon STN during high-frequency stimulation. It is likely that this persistent hyperpolarizing tone is what gives rise to the well-known rebound burst phenomenon that occurs in STN after stimulation is ceased. This is supported by the following scientific findings:

(1) DBS produces suppression of STN neuronal activity *during* HFS (Tai et al., FASEB J 2003) by persistent synaptic inhibition from GPe (Steiner et al., J Neurosci 2019; Steiner et al., Brain Stim 2022) that results in a prolonged hyperpolarizing tone onto STN neurons. Importantly, only strong recurrent inhibition is able to produce complete neuronal suppression (current work, Fig. 2).

(2) Rebound bursts in STN occur in up to 75% of stimulation trains *after* stimulation is stopped (Milosevic et al., JNNP 2019), whereas firing is generally suppressed *during* stimulation.

(3) Mechanistically, rebound bursts are known to occur *after* periods of prolonged hyperpolarization (i.e., deinactivation of T-type calcium channels; Tai et al., J Clin Invest 2011).

Thus, the presence of rebound bursts indicates the presence of a persistent hyperpolarizing drive. Because STN receives inhibitory input almost exclusively from GPe, this hyperpolarizing drive can be expected to be GPe-mediated.

To address the Reviewer's insightful comment, we have implemented a conductance-based model of STN with known membrane properties (L and T type calcium currents, A

currents, and calcium dependent AHP currents), in which we input the current that is produced in our model (tonic hyperpolarization imposed by direct and recurrent activation of GPe afferents; i.e., ERNA). The results of the conductance-based model indeed confirm the aforementioned points that (1) spiking activity is inhibited *during* stimulation-induced currents (i.e., net hyperpolarization), whereas (2) the rebound burst phenomenon occurs *after* stimulation-induced hyperpolarization (3) as a result of the release of inhibitory tone. These *in silico* results bear striking resemblance to empirical data that are provided for reference.

Amendments:

The following has been added to the Discussion (Pg. 13):

“Finally, the hyperpolarizing tone that is produced by ERNA can be expected to underlie the well-known rebound burst phenomenon (that occurs after prolonged hyperpolarization; mediated by deinactivation of T-type calcium channels³⁹) that is observed in STN after stimulation cessation⁸ (as is recapitulated in both in our empirical and *in silico* data; Supplementary Fig. 2).”

We have furthermore added a new figure to the Supplementary Material (see next page), and added the corresponding methodological details and results to the Supplementary Material:

“Supplementary Figure 2 – Methods:

We employed a detailed conductance model of an STN neuron (Hahn and McIntyre, Journal of computational neuroscience 2010) simulated in NEURON¹ to investigate the response characteristics during and after DBS. The STN neuron was injected with fluctuating synaptic conductances to mimic background activity so that the model had an expected baseline firing rate.² The conductance model was simulated receiving the synaptic activity generated from our LIF based model during DBS (i.e., *in silico* ERNA). Note that the model parameters were derived from rats³ instead of humans, which may lead to differences in the time course of responses.

Supplementary Figure 2 – Results:

The conductance model receiving synaptic activity derived from our modelled activation of synaptic afferents (from Fig. 4 of the main manuscript) was able to produce similar post-hyperpolarization rebound bursting activity as shown in representative empiric data. During ongoing DBS, neuronal activity was completely inhibited until the stimulation train terminated, after which rebound bursting activity occurred (both *in silico* and in the representative empiric data).”

Supplementary Figure 2 - Conductance-based model of STN confirms that spiking activity is inhibited during stimulation-induced currents and replicates rebound burst phenomenon after stimulation-induced hyperpolarization. (A) Representative example of a 10 s train of stimulation at 100 Hz (100 μ A; 0.3 ms biphasic pulses). Upper panel shows raw trace with stimulation artefacts removed for clarity. Note that neuronal firing is completely suppressed during ongoing stimulation. Superimposed pink trace represents same recording smoothed with 0.1 s time constant to highlight net inhibitory (stimulation-induced) current produced by stimulation. (B) In silico membrane potential response of the conductance-based model of STN in response to stimulation-induced currents. (Upper panels, in A & B) 13.5 s trains of stimulation / simulation including 1s baseline firing; 10 s of stimulation / stimulation-induced currents and 1.5 s of (in silico) neuronal activity after stimulation. (Lower panels, in A & B) Same traces as in upper panels at higher temporal resolution to highlight after hyperpolarization rebound burst phenomenon in both empirical (A) and in silico data (B).

(2) In the absence of the computational model exhibiting appropriate membrane properties, it becomes difficult to be convinced by the following statement "In particular, the initial peak is conceptualized to be the result of sustained GPe-mediated inhibition to STN, whereas the second peak is the result of synaptic competition at the level of GPe which feeds back to STN to produce recurrent inhibition." which entirely relies on the computational model.

Thank you for raising this very important point. As requested, we have integrated the currents generated by our modelling framework into a computational model exhibiting appropriate membrane properties (see above).

While computational models indeed rely upon assumptions, we would furthermore like to point out that there is empirical evidence to support the statement the Reviewer refers to. DBS-like stimulation in the acutely isolated rat brain slice, that does not contain GPe neurons but only GABAergic axonal blebs, does not produce a second inhibitory peak (Steiner et al, *J Neurosci*, 2019), which is indicative of the fact that the integrity of GPe-STN reciprocal circuit is necessary to produce the phenomenon of recurrent inhibition (i.e., the second inhibitory peak of ERNA) that is highlighted in this paper.

Furthermore, we would refer to the work of Kita and colleagues, who have shown that HFS in STN evoked neuronal excitations in GPe (Kita et al., *J Neurosci*, 2005). The authors provide data to show that these neuronal excitations are entrained, and progressively increase in strength within 10 pulses of 100 Hz, in alignment with the dynamics of the growth of P2 reported on in this paper. This entrainment was abolished after the local applications of glutamate receptor antagonists. Thus, Kita et al. provide evidence of neuronal excitement in GPe that is governed by synaptic effects of the STN efferent projection.

Amendments to the manuscripts:

The following has been added to the Discussion (Pg. 13):

"Kita et al. have shown that synaptically-mediated neuronal excitations in GPe during ongoing STN-HFS progressively increase in strength within 10 pulses of 100 Hz¹⁴ and are thus in good temporal alignment with the emergence of recurrent neuronal inhibition presented in our study (Fig. 3, Fig. 4, and Supplementary Fig. 1)."

We have also added the following (Pg. 14):

"Importantly, our *in silico* work suggests that ERNA depends on the activation of the STN-GPe circuit. While the initial peak of the ERNA waveform can be explained by sustained GPe-mediated inhibition to STN, the second peak is proposed to be the result of synaptic competition at the level of GPe that is dominated by STN efferent activation,³⁹ which ultimately feeds back to STN to produce recurrent inhibition. This dependence on an intact reciprocal network is corroborated by findings that ERNA is not produced by DBS-like stimulation of STN in acutely isolated rat brain slices that do not contain GPe neurons, but only GABAergic axonal blebs.⁹ This suggests that the integrity of GPe-STN circuit is necessary to produce the phenomenon of recurrent inhibition."

The authors would like to emphasize that the reliance on assumptions in the computational modelling framework has been acknowledged within the Limitations (Pg. 16):

"Whenever possible, the physiological components of our computational modelling framework utilized parameters to emulate synaptic dynamics based on empirical data in the present and previous studies. However, to the best of our knowledge, no MER data exists on the dynamics of evoked potentials and corresponding firing in GPe in response to STN stimulation. Because pallidal firing is continuously driven by STN-HFS,^{12,39} we assumed a

resilient STN output synapse. When we changed the STN output synapse to not be resilient *in silico*, the characteristic ERNA waveform did not develop (Fig. 4C). Indeed, this resiliency is supported by intraoperative biochemical studies reporting sustained glutamate levels in SNr during STN-DBS.²⁹”

However, we have now supplemented the Considerations & Limitations section (Pg. 16) with a suggested patch-clamp experiment that may provide more direct evidence of cell-type specific neuronal and synaptic recruitments and their involvement in the emergence of ERNA, as referred to by the reviewer.

“Future simultaneous patch-clamp recordings of GPe neurons during STN stimulation in acute pallidal-subthalamic slices may provide more direct evidence of cell-type specific neuronal and synaptic recruitments and their involvement in the emergence of ERNA.”

REVIEWERS' COMMENTS

Reviewer #1 (Remarks to the Author):

The authors have responded to reviewer concerns. Figure 3D, with the revision and clarification, now yields to my attempts to understand it. one piece of the authors' response comments was the breakthrough moment in understanding the figure: that STN firing itself is actually not necessary for the recurrent inhibition that generates the second ERNA waveform. i realized then that a challenge of this paper is that, recording of STN neuronal firing is used as tool to INFER synaptic inputs and thus to infer the behavior of the synaptic currents that generate ERNA, but is NOT itself needed for the expression of ERNA (because steps 1B and 2 in figure 3D depend on only on neurostimulation activating fibers, not on cell bodies depolarizing to produce an action potentials). However, ERNA that is evoked by STN DBS and measured in STN, DOES require GPe action potential discharge so as to initiate the GPe-to-STN activity in step 2. (unless i still misunderstand the figure). It could help the reader to state these conceptual foundations specifically.

While not important to this specific paper, in looking at the figure from their response to reviewers (copied from their recent review in BRAIN that goes into how GPi stim generates GPi ERNA), it isnt as clear to me whether STN firing is or is not required for pallidal ERNA. it seems that there could be pallidal ERNA mechanisms that are independent of STN cell firing (as the arrows in their reproduced figure imply), but there could be another mechanism that IS depending on STN firing. that may be the topic of a future paper.

Reviewer #1 (Remarks on code availability):

i do not have expertise in computational models

Reviewer #2 (Remarks to the Author):

The authors have addressed all my comments. I believe this body of work will be a valuable addition to the brain stimulation literature.

NCOMMS-23-25757 [R2]

(Neural signatures of indirect pathway activity during subthalamic stimulation in Parkinson's disease)

Responses to Reviewer #1

Reviewer #1 (Remarks to the Author):

The authors have responded to reviewer concerns. Figure 3D, with the revision and clarification, now yields to my attempts to understand it. One piece of the authors' response comments was the breakthrough moment in understanding the figure: that STN firing itself is actually not necessary for the recurrent inhibition that generates the second ERNA waveform. I realized then that a challenge of this paper is that, recording of STN neuronal firing is used as a tool to INFER synaptic inputs and thus to infer the behavior of the synaptic currents that generate ERNA, but is NOT itself needed for the expression of ERNA (because steps 1B and 2 in figure 3D depend only on neurostimulation activating fibers, not on cell bodies depolarizing to produce an action potential). However, ERNA that is evoked by STN DBS and measured in STN, DOES require GPe action potential discharge so as to initiate the GPe-to-STN activity in step 2. (unless I still misunderstand the figure). It could help the reader to state these conceptual foundations specifically.

While not important to this specific paper, in looking at the figure from their response to reviewers (copied from their recent review in BRAIN that goes into how GPi stim generates GPi ERNA), it isn't as clear to me whether STN firing is or is not required for pallidal ERNA. It seems that there could be pallidal ERNA mechanisms that are independent of STN cell firing (as the arrows in their reproduced figure imply), but there could be another mechanism that IS depending on STN firing. That may be the topic of a future paper.

Reviewer #1 (Remarks on code availability):

I do not have expertise in computational models

Reply:

Thank you again for your valuable feedback. The provided interpretation is correct. We have further amended the Fig. 3 legend for clarification. The text below highlights edits to Fig. 3 legend text; however, only a clean copy of the manuscript has been uploaded to the manuscript submission system (without tracked changes).

Amendment:

(1a) depicts direct simultaneous activations of the inputs to STN. Sustained GPe transmission is paired with rapidly depressing cortical transmission, which leads to a sustained net inhibition in STN (this inhibits STN spike firing and produces a hyperpolarization of the membrane potential which is reflected as P1 of STN ERNA, filled grey). (1b) depicts direct activation of STN efferent outputs (even though spike firing is inhibited by GPe activation, STN efferent axons are nevertheless expected to be activated by DBS pulses) and GPe-GPe collaterals (by way of antidromic activation of GPe-STN projections, and subsequent invasion of GPe-GPe collaterals). Sustained GPe transmission paired with even more sustained STN transmission leads to an increasing net excitation in GPe. (2) depicts that the increasing net excitation in GPe feeds back to the STN, producing recurrent inhibition of STN (this produces a second hyperpolarization of the membrane potential in STN within the interstimulus interval, which is reflected as P2 of STN ERNA, filled black). Effectively, the recurring activations of GPe inhibit the generation of STN spike firing, while contributing to the positive-going voltage peak deflections in the ERNA waveform (i.e., due to a loss of negatively-charged ions in the extracellular field potential recordings due to GABAergic activations).

Responses to Reviewer #2:

Reviewer #2 (Remarks to the Author):

The authors have addressed all my comments. I believe this body of work will be a valuable addition to the brain stimulation literature.

Reply:

Thank you again for your valuable feedback.